# New particle formation in the remote marine boundary layer

Guangjie Zheng [1,2,10], Yang Wang [1,3,10], Robert Wood [4], Michael P. Jensen[2], Chongai Kuang[2], Isabel L. McCoy [4], Alyssa Matthews [5], Fan Mei[5], Jason M. Tomlinson [5], John E. Shilling [5], Maria A. Zawadowicz[5], Ewan Crosbie[6,7], Richard Moore [6], Luke Ziemba[6], Meinrat O. Andreae [8,9] & Jian Wang [1,2✉]

Marine low clouds play an important role in the climate system, and their properties are sensitive to cloud condensation nuclei concentrations. While new particle formation represents a major source of cloud condensation nuclei globally, the prevailing view is that new particle formation rarely occurs in remote marine boundary layer over open oceans. Here we present evidence of the regular and frequent occurrence of new particle formation in the upper part of remote marine boundary layer following cold front passages. The new particle formation is facilitated by a combination of efficient removal of existing particles by precipitation, cold air temperatures, vertical transport of reactive gases from the ocean surface, and high actinic fluxes in a broken cloud field. The newly formed particles subsequently grow and contribute substantially to cloud condensation nuclei in the remote marine boundary layer and thereby impact marine low clouds.

[1] Center for Aerosol Science and Engineering, Department of Energy, Environmental and Chemical Engineering, Washington University in St. Louis, St. Louis, MO, USA. [2] Environmental and Climate Science Department, Brookhaven National Laboratory, Upton, NY, USA. [3] Department of Civil, Architectural and Environmental Engineering, Missouri University of Science and Technology, Rolla, MO, USA. [4] Department of Atmospheric Science, University of Washington, Seattle, WA, USA. [5] Atmospheric Measurement & Data Sciences, Pacific Northwest National Laboratory, Richland, WA, USA. [6] NASA Langley Research Center, Hampton, VA, USA. [7] Science Systems and Applications, Inc., Hampton, VA, USA. [8] Max Planck Institute for Chemistry, Mainz, Germany. [9] Scripps Institution of Oceanography, University of California San Diego, La Jolla, CA, USA. [10] These authors contributed equally: Guangjie Zheng, Yang Wang. ✉email: jian@wustl.edu

Marine boundary layer (MBL) clouds have large spatial and temporal coverage and play an important role in the climate system[1]. Substantial variability in their radiative effects is attributed to the concentration of cloud condensation nuclei (CCN)[2,3]. New particle formation (NPF), namely vapor nucleation and subsequent formation of new particles, in the MBL over open oceans has the potential to increase the CCN concentration, thereby influencing clouds and climate[4]. However, despite extensive studies, observational evidence of NPF in the remote MBL over open oceans remains scarce and sometimes ambiguous[5–13]. Nucleation mode particles, which include newly formed particles and new particles that have undergone early growth, have been observed near the ocean surface during previous studies (e.g., research cruises)[7,9,13]. However, they are conventionally attributed to entrainment from the free troposphere (FT)[7,9,14], where NPF is frequent[15–19]. This attribution is largely based on the following results. First, model studies show that NPF in the MBL is unlikely under typical marine conditions[14], because the existing aerosol surface area is large due to primary sea spray aerosols (SSA)[9], and precursor concentrations are low[14]. Second, the concentration of nucleation mode particles observed in the MBL often exhibited no anti-correlation with existing particle surface area, as would be expected during NPF[7]. Therefore, it has been long thought that NPF in the remote MBL is sporadic and occurs only under exceptional conditions[14,20,21], thus rendering MBL NPF a minor factor in the CCN budget. However, long-term observations suggest that the frequently observed nucleation mode particles in the MBL are unlikely to be fully explained by FT entrainment[13].

Here, we present evidence of the regular and frequent occurrence of NPF in the upper part of the mid-latitude MBL, based on airborne measurements and long-term surface-based observations. These measurements show that nucleation mode particles observed near the ocean surface often originate inside the MBL, instead of being entrained from the FT as previously thought. The NPF is made possible by the combination of low existing particle concentrations, cold temperature, availability of reactive gases, and high actinic fluxes in the clear regions between scattered cumulus clouds following the passage of cold fronts. The newly formed particles subsequently grow and contribute substantially to CCN in the MBL, and thereby may influence marine low clouds and climate.

## Results and discussion

**Observations of new particle formation in remote marine boundary layer.** Comprehensive characterizations of aerosols, trace gases, and cloud microphysics were carried out onboard the U.S. Department of Energy Gulfstream-1 (G-1) research aircraft during the Aerosol and Cloud Experiments in the Eastern North Atlantic (ACE-ENA) campaign. Our airborne measurements show elevated concentrations of newly formed particles in the upper part of the remote MBL during conditions of open mesoscale cellular convection or scattered cumulus clouds following the passages of cold fronts. One such example was observed on 16 February 2018 (the synoptic condition is shown in Supplementary Fig. 1). Vertical profiles of potential temperature ($\theta$) and carbon monoxide (CO) mixing ratio (a passive tracer) show a relatively deep MBL with a height of 2000 m (Fig. 1a, b). The profiles of $\theta$ and water vapor mixing ratio indicate that the MBL was decoupled, with a lower surface mixed layer up to ~ 1050 m and an upper decoupled layer from ~ 1050 to 2000 m (Fig. 1a). The ion signal at m/z 63 measured by a Proton-Transfer-Reaction Mass Spectrometer (PTR-MS) suggests that vertical profile of dimethyl sulfide (DMS) is similar to that of water vapor, showing elevated mixing ratio in both surface mixed

layer and upper decoupled layer compared to that in the free troposphere (Fig. 1b). This is expected as DMS originates from the ocean surface. Cumulus clouds were observed at the edge of open cells (Fig. 1d). Aerosol properties shown in this study are based on measurements outside of the clouds to avoid artefacts potentially arising from the shattering of cloud droplets on sampling inlets (see Methods). An abundance of newly formed particles is indicated in the upper decoupled layer by the elevated concentration ratio of particles with diameter larger than 3 nm to particles larger than 10 nm ($N_{>3nm}/N_{>10nm}$) (Fig. 1c). The number fraction of particles volatile at 300 °C is close to 100% in the regions with elevated $N_{>3nm}/N_{>10nm}$ (Supplementary Fig. 2), consistent with the volatile nature of typical incipient particles in the atmosphere[22,23]. While $N_{>3nm}/N_{>10nm}$ is also elevated in the FT at ~3000 m, the absence of particles smaller than 10 nm (i.e., $N_{>3nm}/N_{>10nm} = 1$) immediately above the MBL suggests that the new particles present in the MBL were formed in the upper decoupled layer rather than having been entrained from the FT. The inference of NPF is also supported by the sharp rise of the particle concentrations in the size range below 20 nm at 1700 m (Fig. 1e). The spatial extent of the particle formation in the upper MBL (Fig. 1d) was significant, at least 50 km.

The $N_{>3nm}/N_{>10nm}$ ratio exhibits a strong anti-correlation with total particle surface area concentration ($S_{tot}$), which drops substantially with height from ~30 $\mu m^2\ cm^{-3}$ in the surface mixed layer to ~ 10 $\mu m^2\ cm^{-3}$ in the upper decoupled layer (Fig. 1c). Here we use $S_{tot}$ to represent the condensation sink of nucleation precursors and the coagulation sink of newly formed particles (see Methods). In the surface mixed layer, contributions to $S_{tot}$ are dominated by coarse mode aerosol (i.e., particles with diameter larger than 1 $\mu m$, Supplementary Fig. 3), consistent with sea spray aerosol. Elevated $N_{>3nm}/N_{>10nm}$ in the upper decoupled layer coincided with reduced water vapor mixing ratio and depleted coarse mode particles between clouds (i.e., in the clear center region of open cells, Supplementary Fig. 2), indicating that these new particles formed in regions where existing particles had been largely removed by precipitation associated with cumulus clouds. This precipitation removal led to a substantially lower $S_{tot}$ in the upper MBL than the typical values in the models that suggested NPF inside the MBL is unlikely[14]. The vertical profile of $N_{>3nm}/N_{>10nm}$, and the aerosol size distribution measured at an altitude of 100 m indicate that the new particles formed in the upper decoupled layer were transported downwards, resulting in a $N_{>3nm}/N_{>10nm}$ value of 1.2 in the surface mixed layer. This also explains the lack of anti-correlation between the nucleation mode particle concentration and $S_{tot}$ reported from previous research cruises[7,9], because the nucleation mode particles observed near the ocean surface likely originated from the upper level of the MBL, where $S_{tot}$ is much lower.

Features similar to those shown in Fig. 1 were also observed in the MBL during comparable postfrontal meteorological conditions of ACE-ENA and during the North Atlantic Aerosols and Marine Ecosystems Study (NAAMES) (Supplementary Figs. 4–7), suggesting that NPF occurs in the upper MBL over different geographic area of mid-latitude oceans (i.e., is not specific to the Azores). It is worth noting that during some ACE-ENA flights, although an elevated $N_{>3nm}/N_{>10nm}$ was absent, the nucleation mode particles in the upper decoupled layer exhibited a substantially smaller mode diameter than the aerosol immediately above the MBL, also indicating that the nucleation mode particles grew from new particles formed in the MBL instead of having been entrained from the FT (e.g., Supplementary Fig. 6).

**The mechanism of new particle formation in the remote marine boundary layer.** The dependence of NPF in the MBL on

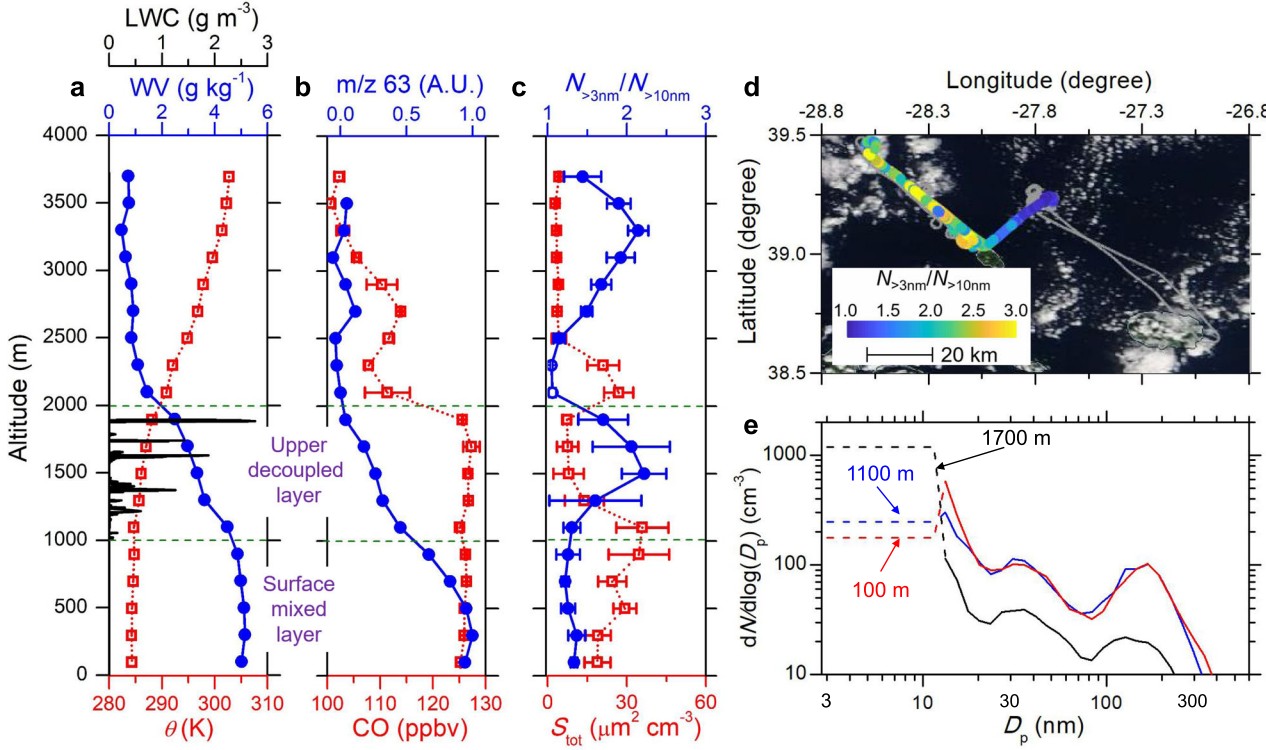

**Fig. 1 Measurements onboard the G-1 aircraft upwind of the Eastern North Atlantic (ENA) site on 16 February 2018. a** Vertical profiles of potential temperature ($\theta$), water vapor mixing ratio (WV, g $H_2O$ / kg wet air), and liquid water content (LWC). **b** Vertical profiles of carbon monoxide (CO) mixing ratio and the ion signal (arbitrary units, A.U.) at m/z 63 (e.g., DMS). The error bars for CO represent one standard deviations of 1-s measurements. **c** Vertical profiles of the concentration ratio of particles larger than 3 nm to particles larger than 10 nm ($N_{>3nm}/N_{>10nm}$) and total particle surface area concentration ($S_{tot}$). Elevated $N_{>3nm}/N_{>10nm}$ (i.e., greater than 1.1) that are statistically significant for the altitude bins (detailed in Methods section) are marked with filled circles and the rest are shown by open circles. The error bars represent one standard deviations for 1 s $N_{>3nm}/N_{>10nm}$ and 10 s $S_{tot}$ values, respectively. **d** Flight track of the G-1 aircraft during horizontal legs in the upper decoupled layer colored by $N_{>3nm}/N_{>10nm}$. During the flight, the wind was from the northeast. The flight tracks include both 30 km along-wind legs upwind of the ENA site and 50 km crosswind legs towards the northwest over the ocean, therefore the potential influence of island sources on G-1 aerosol measurements is negligible. The background image from NASA Worldview, taken by MODIS at an earlier time than the flight on the same day, is used to illustrate the open-cell convection cloud field. **e** Particle size distributions measured at three different altitudes within the marine boundary layer. The particle concentrations below 10 nm (shown by dashed lines) are derived as the difference between $N_{>3nm}$ measured by the CPC and the concentration of particles larger than 10 nm integrated from the FIMS size distribution. The vertical profiles shown in (A) and (B) are based on measurements from 13:28 to 14:00 UTC. The liquid water content during the entire flight is shown to illustrate the vertical extent of clouds. Source data are provided as a Source Data file.

$S_{tot}$ is further examined using measurements from all 39 flights during ACE-ENA (Fig. 2). An elevated $N_{>3nm}/N_{>10nm}$ in the MBL was observed exclusively when $S_{tot}$ was below ~10 $\mu m^2$ $cm^{-3}$, which is comparable to typical values in the FT. However, low $S_{tot}$ alone is not sufficient for MBL NPF to occur; temperature and cloud conditions also play an important role (Fig. 2). Despite low $S_{tot}$, $N_{>3nm}/N_{>10nm}$ remained close to unity during several flights under overcast conditions. The absence of new particles is likely a result of (a) weak production of precursors due to slower photo-oxidation under overcast conditions, and (b) interstitial scavenging of precursors and newly formed particles by droplets inside the cloud layer. In addition, elevated $N_{>3nm}/N_{>10nm}$ was always associated with lower ambient temperature. This is consistent with a previous observation of NPF in the tropical MBL[10], where an abrupt drop of temperature was detected during the NPF event. Recent chamber measurements of atmospheric nucleation rates show that a temperature decrease of 5 K results in 5 to 50-fold increases in the nucleation rates[23].

The above results and the vertical profiles of DMS mixing ratio (e.g., Fig. 1b, Supplementary Fig. 7b) indicate that NPF in the MBL is due to a combination of low $S_{tot}$, cold ambient temperature, availability of the precursors, and photo-oxidation within the clear regions of open-cell convection or a scattered cloud field. These are the typical conditions in the upper decoupled layer following the passage of cold fronts. The underlying mechanism is illustrated in Fig. 3. When cold air behind a front travels across open ocean surfaces, it gradually gains heat and moisture, resulting in a deepening MBL[24]. As the MBL deepens, the turbulence produced from surface-heating and cloud-top radiative cooling becomes insufficient to maintain a well-mixed layer. Consequently, the MBL begins to "decouple" into a surface mixed layer and an upper decoupled layer[25–27]. In a decoupled MBL, the upward transport of particles and trace gases produced in the surface mixed layer to the upper decoupled layer is largely through rising thermals that form cumulus clouds. Because cumulus-associated drizzle and precipitation efficiently remove existing large particles such as SSA, this leads to a much reduced $S_{tot}$ in the clear air between clouds in the upper decoupled layer, as shown in Fig. 1 and reported in earlier studies[28,29]. Meanwhile, reactive gas-phase species of low water solubility such as DMS are not scavenged and subsequently detrained into the upper decoupled layer. In the clear (i.e., cloud-free) region between clouds, these reactive gases undergo more rapid photo-oxidation and produce nucleation precursors, including $H_2SO_4$ and methanesulfonic acid (MSA). A recent study discovered a new oxidation product of DMS,

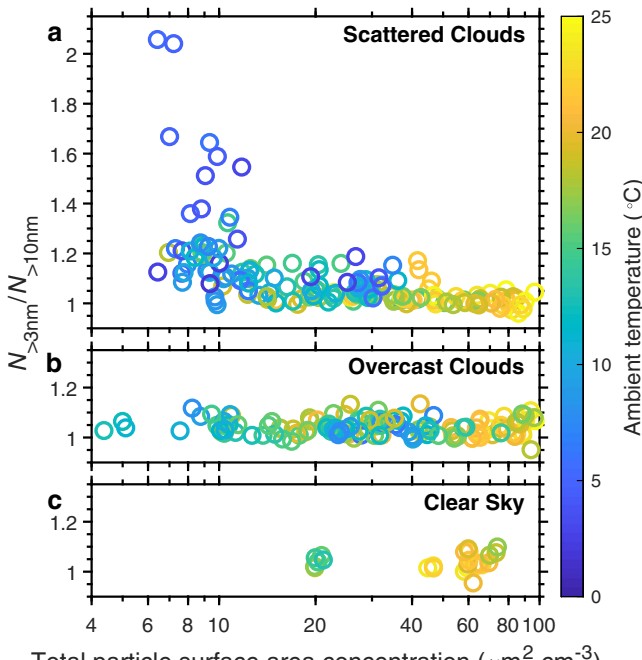

**Fig. 2 Key factors driving new particle formation in the marine boundary layer (as indicated by elevated $N_{>3nm}/N_{>10nm}$).** Dependence of $N_{>3nm}/N_{>10nm}$ on total particle surface area concentrations and ambient temperature is shown under different cloud conditions during the ACE-ENA campaign. These conditions include: **a** scattered clouds (cloud fraction between 0.3 and 0.85); **b** overcast conditions (cloud fraction over 0.85), and **c** clear sky (cloud fraction below 0.3). The cloud fraction is determined from the measurement of a total sky imager at the ENA site. Each data point represents the average of measurements within an altitude bin of 200 m inside the marine boundary layer during the vertical profiles by the G-1 aircraft. Source data are provided as a Source Data file.

hydroperoxymethyl thioformate, which may also contribute to the NPF and particle growth in marine environment[30]. The combination of low existing $S_{tot}$, availability of reactive gases such as DMS, high actinic flux for photo-oxidation in the clear region between clouds, and low ambient temperature makes possible the observed NPF in the upper MBL.

Subsequently, these newly formed particles are transported downwards by clear-air subsidence compensating the cumulus updrafts, and mixed into the surface layer by turbulent mixing at the boundary between the upper decoupled layer and the surface mixed layer. The downward transport of newly formed particles is evident from the vertical profile of $N_{>3nm}/N_{>10nm}$ (Fig. 1c). During the downward transport and subsequent residence in the lower MBL, new particles likely undergo growth and reach larger sizes, as suggested by the shift of the nucleation mode particles towards larger diameters in the surface mixed layer (Fig. 1e). Over the mid-latitude oceans, open-cell convection and scattered cumulus clouds frequently occur behind cold fronts, and satellite images show that their spatial coverage is usually very extensive, on the order of several hundred kilometers. Therefore, it is expected that the NPF occurs in the upper MBL over large geographic area of mid-latitude oceans.

The above airborne measurements indicate that following the passage of cold fronts, nucleation mode particles observed near the ocean surface often originate from NPF in the upper MBL, rather than the FT. Regular and frequent occurrence of MBL NPF is supported by the year-long (June 2017 to June 2018) measurements at the Eastern North Atlantic (ENA) site on Graciosa Island, which show nucleation mode particles and their growth during a total of 67 events, all of which are associated with a decoupled MBL following the passage of a cold front (Methods, Supplementary Fig. 8a). An example of the growth events is shown in Supplementary Fig. 8b. Among the 67 events, 65 events occurred with open cell convection or scattered cumulus clouds (Supplementary Data 1). Low $S_{tot}$ in the upper decoupled layer during these events is supported by Raman lidar measurements

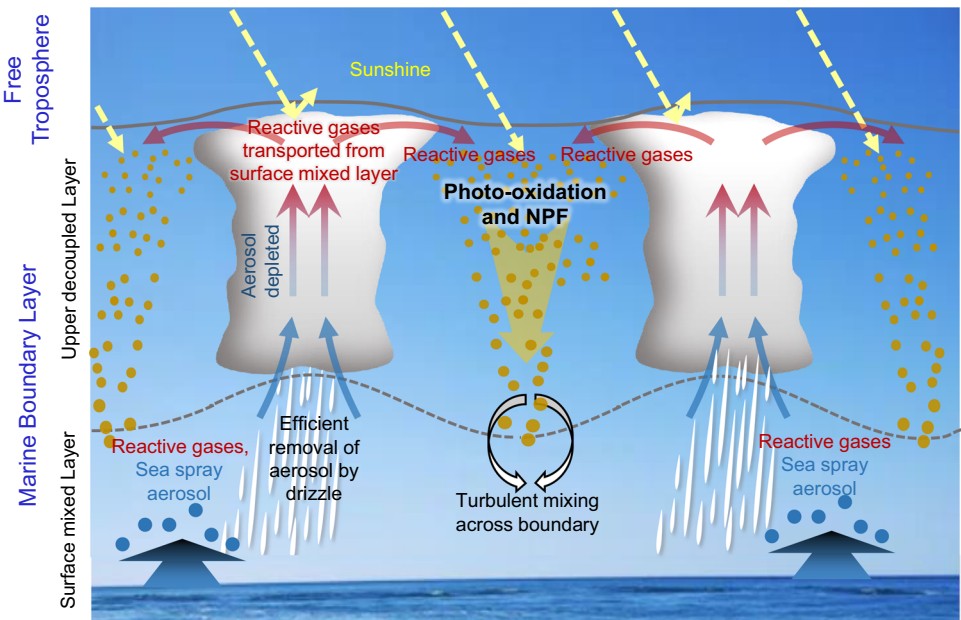

**Fig. 3 Mechanism of new particle formation in a decoupled marine boundary layer when clouds form open cell convection or exhibit a scattered cloud field following the passage of a cold front.** In a decoupled marine boundary layer, particles and ocean-emitted trace gases are transported upwards from the surface mixed layer to the upper decoupled layer through rising thermals that form cumulus clouds. Cumulus-associated drizzle and precipitation efficiently remove large particles and thus reduce existing aerosol surface area, while reactive gases of low water solubility survive and are detrained into the upper decoupled layer. In the clear region between clouds, these reactive gases undergo more rapid photo-oxidation and produce nucleation precursors, leading to new particle formation. The newly formed particles are subsequently mixed down to the surface layer through turbulent mixing.

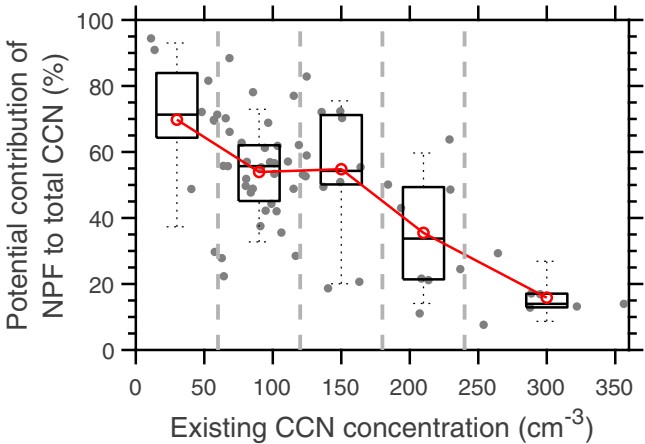

**Fig. 4 Potential contribution of marine boundary layer new particle formation (NPF) to cloud condensation nuclei (CCN) population as a function of existing CCN concentration.** The contribution is calculated as the concentration ratio of CCN originating from new particles formed in the marine boundary layer (i.e., NPF CCN) to total CCN for the 67 events observed at the Eastern North Atlantic (ENA) site. NPF CCN represents the nucleation mode particles that survived coagulation scavenging until reaching the CCN size (i.e., 80 nm), and total CCN is the sum of existing CCN and NPF CCN. The gray dots represent the results for individual events, and the dashed gray lines indicate the boundaries (i.e., 60, 120, 180, and 240 cm$^{-3}$) between existing CCN concentration bins. The numbers of events in the five existing CCN concentration bins are 9, 31, 12, 8, and 7 from low to high concentration, respectively. The box-whisker plot is drawn for 10-, 25-, 50-, 75-, and 90-percentiles. The red circles represent the mean values. Source data are provided as a Source Data file.

(Methods, Supplementary Fig. 9, and Supplementary Data 1). Furthermore, the entrainment of FT aerosol into the MBL is likely too slow to explain the nucleation mode particles and their growth observed during many events (see Methods). All the above lines of evidence support the role of NPF in the upper MBL during these frequently occurring events.

**Contribution of condensational growth of newly formed particles to marine boundary layer cloud condensation nuclei.** The potential contribution of NPF in the upper MBL to the CCN populations was estimated for these 67 events (see Methods). On average, the growth of nucleation mode particles, which originate from the upper decoupled layer, contributes 143 ± 118 cm$^{-3}$, 50% ± 22% of the total CCN concentration. The contribution potentially reaches 70% ± 20% under the cleanest conditions when the pre-existing CCN concentration is below 60 cm$^{-3}$ (Fig. 4), suggesting NPF in the MBL is likely an important source that helps replenish aerosol and CCN populations in the pristine marine environment. Nucleation mode particles were also frequently observed in the remote MBL following the passage of cold fronts during previous research cruises in the mid-latitude Pacific Ocean, and in the southern Pacific and Indian Oceans[7,9]. These nucleation mode particles are conventionally attributed to the entrainment of new particles formed in the FT[7,9,14]. The similar observations of nucleation mode particles in the MBL following the passage of cold fronts in different regions also indicate that the findings presented herein may be applied more generally to mid-latitude oceans. The relative frequency of occurrence of cold fronts is generally 5% to 30% over the mid-latitude oceans[31].

Within the cold air outbreak regions, the occurrence probability of broken clouds (i.e., the conditions for NPF to occur) is ~45%[32]. Together, we estimate the NPF in the upper part of the MBL occurs ~2.5% to ~14% of the time over mid-latitude oceans, consistent

with a frequency of 12% based on the observations at the ENA site (Supplementary Data 1). This frequency has a similar magnitude as the frequency of continental NPF events of 10–30%[33], suggesting that new particle formation in the upper MBL likely plays an important role in the budget of MBL CCN.

In summary, the observations reported here show that following the passage of cold fronts, NPF takes place in the upper part of the decoupled MBL over mid-latitude oceans. The NPF is a result of the combination of low $S_{tot}$, cold temperature, availability of reactive gases, and high actinic fluxes in the clear regions between scattered cumulus clouds. Long-term observations suggest that such NPF occurs regularly and frequently, and that the growth of the new particles helps replenish aerosol and CCN in the remote MBL following the passage of cold fronts. The contribution of MBL NPF to CCN populations is expected to have been more significant during the pre-industrial era when anthropogenic aerosol was minimal and total condensation sink was likely lower. As current uncertainties in model simulations of climate change are in large part a result of our poor understanding of processes under natural conditions, NPF in the MBL and its importance need close attention.

## Methods
### Measurements
*Aerosol and cloud experiments in the Eastern North Atlantic (ACE-ENA) campaign.* The Aerosol and Cloud Experiments in the Eastern North Atlantic (ACE-ENA) campaign was designed to improve the understanding of MBL aerosol processes, cloud and drizzle microphysics, and the impact of aerosol on marine low clouds and precipitation in the Eastern North Atlantic (ENA) region by combining airborne observations and long-term surface based measurements. The ENA is a region of persistent but diverse subtropical MBL clouds, the albedo and precipitation of which are highly susceptible to perturbations in aerosol properties. During ACE-ENA, the G-1 research aircraft of the U.S. Department of Energy Atmospheric Radiation Measurement (ARM) Aerial Facility[34] was deployed in two intensive operational periods (IOPs), one during the summer of 2017 and the other in winter of 2018. Flights were carried out in the vicinity of an atmospheric observatory (i.e., ENA site) on Graciosa Island in the Azores, Portugal (39° 5′ 30″ N, 28° 1′ 32″ W, 30.48 m above mean sea level). The flights were carefully designed to ensure negligible influence on aerosol measurements from local sources on the Island. Graciosa Island has a length of 10 km and a width of 7 km, and the ENA site is about ~ 500 m from the north shore. For flights near the ENA site, the flight tracks include both along-wind legs upwind of the ENA site and crosswind legs over the ocean (e.g., Fig. 1). Therefore, the potential influence of the local sources on the aerosol measurements is negligible and the G-1 measurements are representative of those over the open ocean. The ENA site was established by the ARM Climate Research Facility in October 2013 to provide continuous measurements of atmospheric state, aerosol, clouds, and precipitation. During the ACE-ENA campaign, additional aerosol measurements, including aerosol size distributions with diameter range of 10 to 470 nm, were carried out at the ENA site from June 2017 to June 2018. ACE-ENA airborne and surface-based measurements used in this study are detailed below.

*Airborne measurements during ACE-ENA.* During ACE-ENA, the G-1 research aircraft was deployed in two IOPs, from 21 June to 21 July 2017, and from 19 January to 20 February 2018, respectively. The G-1 was stationed at the Lajes Airport located on Terceira Island in the Azores, and 20 and 19 research flights were carried out during the two IOPs, respectively. The G-1 research aircraft was equipped with instruments for characterizing aerosol particles, cloud droplets, and trace gas species, in addition to those for measuring solar radiation, atmospheric state variables and meteorological and avionics parameters[34].

The mixing ratio of CO was measured by a CO/N$_2$O analyzer (Los Gatos Research, Model 9070015). An Ionicon quadrupole high-sensitivity Proton-Transfer-Reaction Mass Spectrometer (abbreviated as PTR-MS hereinafter) was used to measure the mixing ratios of selected gas-phase VOC including DMS. Because the quadrupole PTR-MS cycled through a number of pre-selected m/z values, the duty cycle for each m/z is limited, leading to a reduced signal to noise ratio. In addition, given the unit mass resolution of the quadrupole PTR-MS, isobaric interference may also lead to a positive bias in the measured DMS mixing ratio. Here we only use the vertical profile of the ion signal at m/z 63 measured onboard the G-1 to qualitatively infer the relative abundance of DMS in the upper part of MBL. The conclusion of this study does not depend on the absolute values of the DMS mixing ratio. Aerosol was sampled through a forward-facing inlet (Brechtel Manufacturing Inc.) with a two-stage diffuser that effectively transmits particles with aerodynamic diameter less than 2.5 μm. The inlet has a first diffuser

tip diameter of 8.0 mm and a second diffuser tip diameter of 13.8 mm. The inlet system is controlled by a feedback loop to isokinetically sample at typical aircraft airspeeds. Measurements used in this study include the aerosol size distribution ranging from 10 to 500 nm in diameter characterized by a fast integrated mobility spectrometer (FIMS)[35–37], and from 0.1 to 3 μm in diameter characterized by a passive cavity aerosol spectrometer probe (PCASP). Both measurements were carried out with a time resolution of 1 Hz. The relative humidity of the aerosol sample was reduced to below ~30% using a Nafion dryer before being introduced into the FIMS and the deicing heater of the PCASP remained on during all flights; therefore, the measured size distributions represented that of the dry aerosol particles. The FIMS periodically sampled downstream of a thermal denuder operated at a temperature of 300 °C to characterize the particle size distribution of non-volatile components. The total number concentrations of particles with diameter nominally greater than ~3 and ~10 nm were measured using two condensation particle counters (CPCs, Model 3025 A and 3772, TSI Inc., Shoreview, MN, USA). The number concentration of nonvolatile particles was characterized by a CPC (Model 3010 TSI Inc.) downstream of the thermal denuder. A single particle soot photometer (SP2, DMT Inc., Boulder, CO, USA) was deployed to quantify black carbon (BC) mass in individual particles. The aerosol size distributions, particle number concentrations, and BC mass concentrations are normalized to standard temperature and pressure (273.15 K and 101.325 kPa). The cloud droplets and drizzle drops were characterized by a fast cloud droplet probe (FCDP, SPEC Inc.) from 1 to 50 μm and a 2D-S (Stereo) probe (SPEC Inc.) from 10 to 1280 μm, respectively.

*Aerosol measurements at the ENA site during ACE-ENA.* Since October 2013, aerosol size distribution ranging from 70 to 1000 nm has been measured by an Ultra-High Sensitivity Aerosol Spectrometer (UHSAS, DMT Inc.) at the ARM ENA site. During the ACE-ENA campaign (June 2017 to June 2018), additional measurements were conducted at the ENA site and are described below. Aerosol size distribution from 10 to 470 nm was characterized by a scanning mobility particle analyzer (SMPS, Model 3938, TSI Inc.). The aerosol number concentration (CN) was measured concurrently by a standalone CPC (Model 3772, TSI Inc.) with a 50% cut-off size of 10 nm. The differential mobility analyzer of the SMPS was operated with sheath and sample flow rates of 5 and 1 L min⁻¹, respectively. Both the SMPS aerosol size distribution and number concentration measurements were alternated between ambient samples and those processed by a thermal denuder operated at 300 °C every 4 min. The relative humidity of the aerosol sample was reduced to below 25% using a Nafion dryer before being introduced into the SMPS or the thermal denuder; therefore, the measured size distributions represented those of the dry aerosol particles or thermally denuded dry particles.

*Airborne measurements during the North Atlantic Aerosols and Marine Ecosystems Study (NAAMES).* During the NAAMES, aerosol particles, cloud droplets, and trace gas species were characterized onboard the N439NA NASA Wallops Flight Facility C-130 aircraft based in St. John's, Newfoundland, Canada[38]. Aerosol was sampled through a forward-facing shrouded solid diffuser inlet that effectively transmits particles with aerodynamic diameter less than 5.0 μm. The inlet has a tip diameter of 6.35 mm and is manually controlled to isokinetically sample at typical aircraft airspeeds. Particle number concentrations were measured with 3-nm and 10-nm nominal 50% lower size cuts using Model 3025 A and 3772 CPC (TSI Inc.), respectively. Size distributions from 10 to 300 nm diameter were characterized with a time resolution of 45 s using a custom-built SMPS consisting of a Model 3081 differential mobility analyzer and Model 3772 CPC (TSI Inc.) operated at 10 L min⁻¹ sheath and 1.0 L min⁻¹ sample flow rates. A 0.5 L lag chamber was installed upstream of the SMPS to dampen fast concentration changes in flight. Data are re-binned to 20 channels per decade. Size distributions in the ranges of 100–3160 nm and 562 nm–5.0 μm were measured by a Laser Aerosol Spectrometer and an Aerodynamic Particle Sizer (TSI Inc.), respectively. All concentrations are reported at standard temperature (273.15 K) and pressure (101.325 kPa). Cloud liquid water content is derived by integrating 2–50 μm droplet size distributions measured by a Cloud Droplet Probe (DMT Inc.) mounted just ahead of the aircraft wing leading edge. Gas-phase dimethyl sulfide mixing ratios were characterized using a Proton-Transfer-Reaction Time-of-Flight Mass Spectrometer (PTR-ToF-MS).

**Condensation and coagulation sinks.** In this study, the total surface area concentration ($S_{tot}$), including that of particles and possible hydrometeors, generally shows strong linear relationships with the condensational rate of $H_2SO_4$ molecules and the coagulation rate of 3 nm particles, in agreement with the results from Williamson et al.[19]. Therefore, $S_{tot}$ is used here as a surrogate for both the condensational sink of nucleation precursors and coagulation sink of newly formed particles. For measurements onboard G-1 during ACE-ENA, $S_{tot}$ was calculated by integrating the combined surface area-size distributions measured by the FIMS, PCASP, and FCDP from 10 nm to 50 μm (Supplementary Fig. 3) assuming spherical particles. As we do not have aerosol size distribution below 10 nm, the upper limit of the contribution from particles between 3 and 10 nm to the coagulation sink is calculated by treating all particles between 3 and 10 nm with a uniform diameter of 10 nm. During the entire ACE-ENA campaign, the upper limit of this contribution is 0.2 ± 0.8%. During the NPF cases shown in this manuscript, the contribution is 0.8 ± 1.5%, with a maximum value of 6.25%, which

occurred inside the NPF region. For the NAAMES campaign, due to the low time resolution of the SMPS measurements, the vertical profile of $S_{tot}$ is derived using the size distribution from 100 nm to 5 μm measured by the Laser Aerosol Spectrometer and Aerodynamic Particle Sizer. Using the average size distribution measured by the SMPS, we estimated that the contribution of particles below 100 nm to the total coagulation sink is 14.9 ± 3.3% in the NPF region. For this study, the coagulation sink is largely dominated by aerosols above 100 nm due to the relative abundance of accumulation-mode and sea spray aerosols in the relevant altitude range (i.e., below ~3000 m).

**Observation of new particle formation inside the MBL using airborne measurements.** Aerosol properties shown in this study are based on measurements outside of the clouds to avoid artifacts potentially arising from the shattering of cloud droplets on sampling inlets. The in-cloud time periods are identified based on the liquid water content derived from droplet spectra measured by the FCDP deployed onboard G-1 during ACE-ENA or the cloud droplet probe deployed on C-130 during NAAMES. A threshold of 0.001 g m⁻³ was used to identify the periods of sampling inside clouds. In addition, measurements collected during, two seconds before, and two seconds after the in-cloud sample periods were excluded to eliminate the possible impact of cloud droplet shattering on the particle measurements.

Newly formed particles lead to an elevated concentration ratio of particles with diameter above 3 nm ($N_{>3nm}$) and 10 nm ($N_{>10nm}$), measured by the CPC 3025 A and CPC 3772, respectively. To identify NPF events, we first group the 1-s measurements into 10 s intervals, which correspond to a spatial scale of 1 km. For each 10-second interval, the ratio of average $N_{>3nm}$ to average $N_{>10nm}$ ($R$) and the corresponding uncertainty of the ratio ($\sigma_R$) are derived. A NPF event is identified when the ratio $N_{>3nm}/N_{>10nm}$ is statistically significant:

$$N_{>3nm}/N_{>10nm} > 1 + 3\sigma_R \tag{1}$$

$\sigma_R$ over a sampling period of $\Delta t$ is written as:

$$\sigma_R = \sigma\left(\frac{N_{>3nm,\Delta t}}{N_{>10nm,\Delta t}}\right) \tag{2}$$

where $N_{>3nm,\,\Delta t}$ and $N_{>10nm,\,\Delta t}$ are average $N_{>3nm}$ and $N_{>10nm}$ over the time interval of $\Delta t$. Based on uncertainty propagation, $\sigma_R$ is given by:

$$\sigma_R = \frac{N_{>3nm,\Delta t}}{N_{>10nm,\Delta t}}\sqrt{\left(\frac{\sigma\left(N_{>3nm,\Delta t}\right)}{N_{>3nm,\Delta t}}\right)^2 + \left(\frac{\sigma\left(N_{>10nm,\Delta t}\right)}{N_{>10nm,\Delta t}}\right)^2} \tag{3}$$

For CPC measurements, the particle concentration ($N_{\Delta t}$) is based on $N_{\Delta t} = C_{\Delta t}/(Q\Delta t)$, where $C_{\Delta t}$ is the number of particle counts measured over the time period of $\Delta t$ and $Q$ is the aerosol sample flow rate of the CPC. We assume that the particle counts are described by Poisson statistics, such that $\sigma(C_{\Delta t}) = \sqrt{C_{\Delta t}}$ and $\sigma(N_{\Delta t}) = \sigma(C_{\Delta t})/(Q\Delta t) = \sqrt{C_{\Delta t}}/(Q\Delta t)$. Therefore, we can derive that

$$\sigma^2(N_{\Delta t}) = \frac{C_{\Delta t}}{(Q\Delta t)^2} = \frac{N_{\Delta t}}{Q\Delta t} \tag{4}$$

Combining Eqs. (3) and (4), we have:

$$\sigma_R = \frac{N_{>3\,nm,\Delta t}}{N_{>10\,nm,\Delta t}}\sqrt{\frac{1}{C_{>3\,nm,\Delta t}} + \frac{1}{C_{>10\,nm,\Delta t}}} \tag{5}$$

A NPF event is identified when $N_{>3nm}/N_{>10nm}$ over a 10-s interval is statistically above 1 (i.e., $N_{>3nm}/N_{>10nm} > 1 + 3\sigma_R$). The analysis is focused on periods with BC mass concentration below 2 ng m⁻³ to exclude the occasional impact of local pollution from the islands.

We note that the measured $N_{>3nm}$ and $N_{>10nm}$, and therefore the derived ratio $N_{>3nm}/N_{>10nm}$ may be impacted by flow instability due to sampling at different altitudes and the variation in the CPC 50% cut-off sizes due to the changes in the altitude (i.e., sampling pressure), particle concentration, and composition. In this study, the NPF in the upper part of the MBL were observed below the altitude of approximately 2000 m, corresponding to a pressure range of 80 to 101 kPa. Over this relatively narrow pressure range, the flow instability of the CPCs is negligible, and the variation in cut-off sizes for the CPC 3772 and CPC 3025 A is also expected to be very minor. The sample flow rate of the CPC 3772 was controlled via a critical orifice and is expected to be constant over the narrow sampling pressure range. Whereas the sample flow rate of the CPC 3025 A is not actively controlled, during the periods when newly formed particles were absent based on the spectral shape of the size distribution measured by the FIMS, $N_{>3nm}/N_{>10nm}$ did not change appreciably with sampling altitude, indicating flow-rate stability in the CPC 3025 A over the relevant sampling altitude range. Although CPC size cut-offs tend to increase with increasing concentration of the sampled aerosol particles, aerosol number concentrations during this study were less than ~5000 cm⁻³, substantially below the concentration thresholds above which the impact on cut-off sizes is expected. In addition, as both CPCs use the same working fluid (butanol), we do not expect that the particle composition-dependence of cut-off sizes strongly impacts the interpretation of $N_{>3nm}/N_{>10nm}$.

Figure 1 in the main text shows that on 16 February 2018, $N_{>3nm}/N_{>10nm}$ was elevated in the upper decoupled layer where $S_{tot}$ was low, indicting newly formed particles. The elevated $N_{>3nm}/N_{>10nm}$ ratio for the altitude bins in the MBL is statistically significant based on the criteria described by Eq. (1). Similar features were also observed during flights on 21 January 2018, 24 January 2018 (Supplementary Fig. 4), 28 January 2018 (Supplementary Fig. 5), 29 January 2018, and 15 February 2018 following the passage of cold fronts. During these flights, newly formed particles were observed in the upper decoupled layer but were absent in the lower FT immediately above the inversion. On 8 February 2018 (Supplementary Fig. 6) and 18 February 2018, while the increase of $N_{>3nm}/N_{>10nm}$ was minor (i.e., up to ~1.15), nucleation mode particles in the upper decoupled layer exhibited a substantially smaller mode diameter than the aerosol just above the inversion. These vertical profiles indicate that following the passage of a cold front, NPF occurs in the upper MBL and the nucleation mode particles observed inside the MBL mostly originate from the upper part of the MBL instead of having been entrained from the FT.

Airborne measurements during NAAMES also provide evidence of NPF inside the MBL (Supplementary Fig. 7). From 16 to 20 September 2017, conditions in the North Atlantic and Labrador Sea were dominated by a cold air outbreak event driven by a surface low pressure area over and just east of Greenland as well as high pressure over eastern Canada. On 19 September, the polar low was located approximately halfway between Iceland and the southern tip of Greenland, and the high was moving slowly east-southeast off the coast of Newfoundland. Visible satellite images showed primarily closed-cell stratocumulus clouds located north of 53°N and west of 41°W, while open cells dominated the rest of the basin up to a remnant cold front stretching from Nova Scotia to just west of Ireland. Vertical profiles of potential temperature $\theta$, water vapor mixing ratio, and DMS mixing ratio indicate a decoupled MBL with a height of ~ 2300 m over this region. The surface mixed layer reached ~900 m and the upper decoupled layer ranged from ~900 to ~2300 m. The ratio $N_{>3nm}/N_{>10nm}$ was as high as 3 in the upper decoupled layer, where $S_{tot}$ was less than 10 $\mu m^2$ $cm^{-3}$. The ratio $N_{>3nm}/N_{>10nm}$ was elevated over a spatial scale of 600 km, suggesting recent NPF in the upper decoupled layer over a large geographic area.

**Evidence of frequent new particle formation in the MBL based on long-term observations at the ENA site**

*Growth of nucleation mode particles observed at the ENA site.* During the one-year ACE-ENA campaign from June 2017 to June 2018, measurements at the ENA site show a total of 67 events during which nucleation mode particles with initial mode diameter below 20 nm exhibited continuous growth over a period of at least 5 h. All nucleation mode growth events occurred following the passage of a cold front. The time of these events and the corresponding cold front passages is illustrated in Supplementary Fig. 8a. An example of the growth events is shown in Supplementary Fig. 8b (see more discussion in "Contribution of condensational growth of newly formed particles to MBL CCN" section). The cold fronts are identified from the Modern-Era Retrospective Analysis for Research and Applications, version 2 (MERRA-2) reanalysis data[39]. The passage of a cold front over the ENA site is evident from the presence of a comma-shaped precipitation belt associated with the cold front, and the 850-hPa equivalent potential temperature, $\theta_E$, that exhibits a strong gradient across the front. During all 67 events, the MBL was decoupled, consisting of a surface mixed layer and an upper decoupled layer. Broken clouds or clouds with open cell organization were observed during 65 of the 67 events (Supplementary Data 1).

The nucleation mode particles during these growth events were not from local sources. Instead, the growth events are regional phenomena based on the following lines of evidence. First, the continuous growth of nucleation mode particles over periods of several hours or more has been attributed to regional-scale new particle formation events[22] by many earlier studies in both clean[40–43] and polluted environments[44,45]. For most of the events observed at the ENA site, continuous growth of nucleation mode particles was observed despite substantial shift in wind direction, suggesting regional scale events. As these regional-scale events occur over relatively large spatial scales, they are unlikely due to local point sources. The relatively large scale of these new particle formation events is supported by the aircraft measurements (Fig. 1d and Supplementary Fig. 7c). Second, the nucleation mode particles are also unlikely due to local sources on Graciosa Island. Early studies show that new particles may form from iodine oxides in coastal regions. Such NPF coincides with low tide in the presence of solar radiation[46]. For the growth events observed at the ENA site, the appearance of the nucleation mode particles does not correlate with low tide, indicating a different mechanism. For example, nucleation mode particles were first observed at ~ 15:00 UTC on 16 February 2018, coinciding with the high tide on that day. In addition, the observed particle growth rates between 10 and 35 nm were mostly 1 nm h$^{-1}$ or lower (Supplementary Data 1). Earlier studies[47–50] showed that particle growth rate decreases with decreasing particle diameter below ~ 40 nm, likely due to the stronger Kelvin effect of smaller particles. Therefore, the growth rates below 10 nm are expected to be even lower than 1 nm h$^{-1}$. The attribution of the nucleation mode particles to NPF over open ocean is further based on the fact that this slow growth rate is insufficient to grow particles from 1 nm to several tens of nanometers in less than 1–2 h (i.e., the maximum transit time from the shore/tidal region to the ENA site). Third, the nucleation mode particles and particle growth

are also unlikely due to emissions from surrounding Azores islands, as evidenced by the analysis of air mass backward trajectories. We calculated 3-day back-trajectories of air masses arriving at 100 m above the ground level at the ENA site, for each hour during the observed events. For over 50% of the events, no trajectories intersected with any of the Azores islands (Supplementary Fig. 10a). For only 12% of the events, more than 50% of the hourly air mass trajectories during the event passed over one or more of the Azores islands (Supplementary Fig. 10a). However, the particle growth rate exhibits no dependence on the fraction of hourly trajectories that passed over the Azores islands during the event (Supplementary Fig. 10b), indicating that observed nucleation mode particles and particle growth are unlikely due to biogenic emissions or anthropogenic activities on surrounding Azores islands.

*Sources of nucleation mode particles inside the MBL.* While there have been observations of nucleation mode particles inside the MBL previously[6–13], these nucleation mode particles are conventionally attributed to the entrainment of new particles formed in the FT[7,9,14]. However, the vertical profiles presented above indicate that nucleation mode particles observed in the MBL were likely formed in the upper part of the MBL. For the growth events outside of the IOPs, when only surface sampling is available, it is more challenging to pinpoint the origin of nucleation mode particles observed at the ENA site. Supplementary Fig. 9 shows the vertical profile of the aerosol extinction coefficient on 16 February 2018 retrieved from Raman Lidar measurement at the ENA site. The aerosol extinction decreased with height from the top of surface mixed layer into the upper decoupled layer, in agreement with the clean layer with reduced surface area concentration at ~1500 m observed onboard the G-1. For all 35 growth events when Raman Lidar measurements are available and with signals above the detection limit (Supplementary Data 1), the retrieved vertical profiles show reduced aerosol extinction in the upper decoupled layer, indicating low surface area concentration that is favorable to NPF, as shown in the cases presented earlier. For decoupled MBLs, the aerosol extinction in the upper decoupled layer on event days has a median value of 0.04 km$^{-1}$, and is generally lower than that of non-event days (median value of 0.05 km$^{-1}$), indicating reduced condensation and coagulation sinks. In addition, extinction in the decoupled upper layer is generally lower than that in the upper 1/3 layer of a well-mixed MBL (median value of 0.07 km$^{-1}$).

Consideration of the time scale required for aerosol subsiding from the FT to populate the MBL also sheds light on the potential source of nucleation mode particles inside the MBL. The rate at which FT air mass is incorporated into the MBL, i.e., the entrainment velocity $w_e$, can be estimated from the budget of the boundary height $z_i$ (e.g., refs. [51,52]). The time variation of $z_i$ is described by:

$$\frac{\partial z_i}{\partial t} + \mathbf{V_h} \cdot \nabla z_i = w_e - w_s \tag{6}$$

where $w_s$ is the large-scale subsidence rate, $\mathbf{V_h}$ is the horizontal wind vector, and $\mathbf{V_h} \cdot \nabla z_i$ represents the variation of boundary layer height due to horizontal advection. Assuming steady state conditions for boundary layer height (i.e., $\partial z_i/\partial t = 0$), we have:

$$w_e = w_s + \mathbf{V_h} \cdot \nabla z_i \tag{7}$$

The NPF events observed at the ENA site occurred during postfrontal conditions typical of open mesoscale cellular convection (MCC) conditions[53,54]. In the 40° latitude range (i.e., the latitude of the ENA site), McCoy et al. show that one standard deviation range of lower tropospheric subsidence rates at 700 hPa for open MCC is 10–20 mm s$^{-1}$. Assuming lower tropospheric divergence is constant with height (consistent with previous analyses, e.g., ref. [55]), the mean subsidence rates at the top of the MBL (typically ~ 2 km for postfrontal MBL) will be in the range of ~ 6.5–13 mm s$^{-1}$. The advection term $\mathbf{V_h} \cdot \nabla z_i$ takes into account the deepening of the MBL as the air flows over increasingly warmer water in postfrontal cases. This term is more difficult to estimate in general. For closed cell regions (e.g. SE Pacific, NE Pacific), the advection term is roughly about 1/3 of $w_s$ (ref. [26]). A rough estimate of the advection term can be made based on a mean wind speed of ~10 m s$^{-1}$ and a gradient in MBL depth of about 1 km per 2000 km of advection around the west side of the cyclone. This gives a value of 10× (1/2000) = 0.005 m s$^{-1}$ (i.e., 5 mm s$^{-1}$). The entrainment velocity at the top of the MBL is therefore estimated using Eq. (2) as ~ 12–18 mm s$^{-1}$ for postfrontal open cells. For a representative postfrontal MBL height of ~ 2 km, the time scale for the entrainment of FT particles is therefore ~ 30–45 h. Such a time scale is longer than the duration of many nucleation mode growth events observed at the ENA site, suggesting that entrainment of FT aerosol is likely too slow to explain the nucleation mode particles observed at the ENA site and their continuous growth. This long timescale also explains the observations during several G-1 flights (25 January, 26 January, and 11 February 2018) that show newly formed particles in the lower FT but absent in the MBL, and the lack of detectable nucleation mode particles near the ocean surface at the ENA site. Given the uncertainty of the $w_e$ estimate, it is possible that entrainment of FT aerosol could contribute to nucleation mode particles observed at the ENA site. Nevertheless, the vertical profiles from airborne measurements during the two campaigns, the presence of a clean layer aloft during the growth events, and the long time scale associated with the FT entrainment suggest that the majority of nucleation mode particles observed during the growth events originated from NPF in the upper part of the MBL, which occurs regularly and frequently following the passage of a cold front.

**Contribution of condensational growth of newly formed particles to MBL CCN.** The growth of nucleation mode particles observed at the ENA site suggests new particles formed in the upper part of the MBL may serve as a substantial source of CCN following the passage of cold fronts, when CCN concentrations are typically low due to wet scavenging. The contribution of the grown new particles to the MBL CCN population is estimated from the observed particle growth rate during these events. As shown from satellite images, the spatial coverage of open cell convection or scattered cloud fields behind a cold front is typically very extensive, on the order of several hundred kilometers. The formation of new particles and their growth in the MBL are likely regional phenomena, as supported by the measurements onboard the G-1 and C-130 (e.g., Fig. 1 and Supplementary Fig. 7), and the observations at the ENA site (see discussion in Section "Growth of nucleation mode particles observed at the ENA site"). Therefore, to the first order, we assume that the apparent particle growth rate observed at the ENA site during these events approximates the actual particle growth rate.

It is worth noting that we did not directly observe the growth of particle mode diameter to CCN size range or the average Hoppel minimum (e.g., Supplementary Fig. 8b) because cloud processing strongly modifies the shape of the aerosol size distribution in the MBL. At the ENA site, the Hoppel minimum is typically around 80 nm based on the annual average[56]. During the post-cold-frontal periods when nucleation mode particles and particle growth were observed (Supplementary Table 1), the Hoppel minimum averaged $64 \pm 7$ nm. The Hoppel minimum represents the "average" threshold size of CCN. Updraft velocity for cloud formation inside the MBL exhibits a range of values. Particles with diameters below the Hoppel minimum can activate and become cloud droplets in stronger-than-average updrafts. Once activated, in-cloud production of sulfate and organics increases the amount of the solute in the droplets more efficiently than condensational growth. If these cloud droplets are not removed by wet scavenging, they can return to particles with increased diameters above the Hoppel minimum upon evaporation, and readily serve as CCN during the subsequent cloud formation. This process leads to the typical bimodal aerosol size distribution in the MBL[57]. Martin et al.[58] estimated the peak supersaturation (i.e., supersaturation near cloud base where droplets are activated) of mid-latitude stratocumulus clouds using droplet closure, and the results show a considerable fraction of the peak supersaturations greater than 0.4%, and the peak supersaturation can reach as high as 0.7%. For ammonium sulfate particles with a $\kappa$ value of 0.61, particles as small as 50 nm can be activated into cloud droplets under a supersaturation of 0.4%. Therefore, while some of the particles grow and reach the Hoppel minimum, we do not observe the growth of the *nucleation mode size* to the Hoppel minimum because (1) a substantial fraction of the particles smaller than the Hoppel minimum are activated in stronger-than-average updrafts and subsequently "jump" to the accumulation mode as a result of cloud processing, and (2) the particle mode size is expected to be substantially below the Hoppel minimum, which represents the minimum between the nucleation/Aitken and accumulation modes.

The contribution of condensational growth of newly formed particles (CCN$_{Nu}$) to the CCN population is simulated for each of the 67 growth events. The value of CCN$_{Nu}$ is derived as the number of nucleation mode particles that survive coagulation scavenging and reach the threshold diameter of CCN (i.e., Hoppel minimum of 80 nm) by condensational growth. As particles below the Hoppel minimum can activate in stronger-than-average updrafts, the simulated values represent lower limit of CCN$_{Nu}$. For the simulation of condensational growth, the particle growth rate (GR) is derived from the size distribution measurement using the following approach. The GR is a function of particle diameter $D_p$ (ref. [59]):

$$GR(D_p) = \frac{dD_p}{dt} = \frac{4D_i M_i}{RT\rho_p} \cdot \frac{f(Kn, \alpha)}{D_p} \cdot (p_i - p_{eq,i}) \quad (8a)$$

where

$$f(Kn, \alpha) = \frac{0.75\alpha(1 + Kn)}{Kn^2 + (1 + 0.283\alpha)Kn + 0.75\alpha} \quad (8b)$$

where $D_i$ is the mass diffusivity of condensate $i$ in air, $M_i$ is its molecular weight, $R$ is the gas constant, $T$ is temperature in K, $\rho_p$ is the particle density. $p_i$ and $p_{eq,i}$ are the vapor pressures of condensate $i$ in the bulk gas-phase and at the aerosol surface, respectively. For a non-volatile condensate like $H_2SO_4$, $p_{eq,i}$ is zero[60]. $Kn$ is the Knudsen number defined as $2\lambda_{mfp}/D_p$, where $\lambda_{mfp}$ is the mean free path. The term $f(Kn, \alpha)$ is the correction due to non-continuum effects (depending on $Kn$) and imperfect surface accommodation (depending on mass accommodation coefficient $\alpha$), and is estimated by the Fuchs-Sutugin approach[59].

Equation 8a shows that at the same condensate vapor pressure, the size dependence of GR originates from the term $f(Kn, \alpha)D_p^{-1}$. As this size dependence is negligible below 35 nm, the initial growth rate is first derived from a linear fit of the nucleation mode diameter ($D_{p,n}$) with time, until $D_{p,n}$ reaches 35 nm or in some cases, the growth is interrupted by air mass change. The rate of subsequent particle growth, $GR(D_p)$, from ~ 35 nm to 80 nm is then calculated assuming the same vapor pressure of nonvolatile condensing species (e.g., sulfuric acid or condensable organics) with the size dependence of condensational growth (i.e., $f(Kn, \alpha)D_p^{-1}$) taken into account.

To simulate the coagulation scavenging of growing nucleation mode particles, we construct an event-average aerosol size distribution from 10 nm to 10 μm using

the following approach. For each growth event, measurements from the SMPS and UHSAS are first combined and averaged to provide size distribution from 10 to 1000 nm. The size distribution from 0.4 to 1 μm is then extrapolated to 10 μm using the spectrum shape of the SSA source function, such that coagulation with coarse mode SSA is also taken into account. The underlying assumption is that inside the MBL, particles with diameters greater than 400 nm are dominated by SSA[56].

The value of CCN$_{Nu}$ is derived for each event using the particle growth rate and aerosol size distribution described above. For each time step $\Delta t$ (1 s in this study), the increase of modal diameter $D_{p,n}$ is given by $GR(D_p) \cdot \Delta t$, while the number of growing particles scavenged by coagulation is derived from the coagulation coefficients calculated using $D_{p,n}$ at that time step and the event-average size distribution. The simulation continues until $D_{p,n}$ reaches 80 nm. For the 67 events, the average CCN$_{Nu}$ is $143 \pm 118$ cm$^{-3}$, and the average contribution to total CCN concentration (CCN$_T$) is $50\% \pm 22\%$ (i.e., average ratio of CCN$_{Nu}$ to CCN$_T$, where CCN$_T$ is the sum of CCN$_{Nu}$ and the concentration of CCN prior to the growth event). The contribution reaches 70% $\pm$ 20% under the cleanest conditions when the pre-existing CCN concentration is below 60 cm$^{-3}$ (Fig. 4), suggesting NPF in the upper MBL is an important source that helps replenish aerosol and CCN in the remote MBL following the passage of cold fronts, when CCN concentrations are low as a result of wet scavenging. The NPF in the upper part of the MBL is expected to have played a more important role in controlling the CCN populations in the pre-industrial era, when the impact from anthropogenic emissions was minimal.

## Data availability
All data used in this study are available at https://www.arm.gov/research/campaigns/aaf2017ace-ena and https://www-air.larc.nasa.gov/cgi-bin/ArcView/naames.2017. Source data are provided with this paper.

## Code availability
The MATLAB codes used to analyze data are available upon reasonable request.

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

## Acknowledgements

We acknowledge the Atmospheric Radiation Measurement (ARM) Climate Research Facility, a user facility of the United States Department of Energy (US DOE), Office of Science, sponsored by the Office of Biological and Environmental Research. Funding was obtained from the Atmospheric System Research (ASR) program (Office of Biological and Environmental Research of US DOE, under contract DE-SC0020259 and DE-AC02-98CH10886) and NASA NAAMES project within the Earth Venture Suborbital-2 Program (NNH13ZDA001N-EVS2). We thank Dr. Tamara Pinterich for her help in the preparation and deployment of the FIMS onboard the G-1 aircraft during ACE-ENA, and staff responsible for the operation and maintenance of the ENA site, especially Prof. Eduardo Azevado, Carlos Sousa, Tercio Silva, and Paul Ortega. We acknowledge the use of imagery from the NASA Worldview application, part of the NASA Earth Observing System Data and Information System (EOSDIS).

## Author contributions

J.W., G.Z., Y.W. and R.W. conceived the project and designed the research. J.W., Y.W., G.Z., C.K., A.M., F.M., J.M.T., J.E.S., M.A.Z., E.C., R.M., and L.Z. carried out the measurements. Y.W., G.Z., and J.W. led the analyses, and J.W., G.Z., and Y.W. led the writing, with major input from R.W., M.O.A., and M.P.J. and further input from C.K., I.L.M., A.M., F.M., J.M.T., J.E.S., M.A.Z., E.C., R.M., and L.Z..

## Competing interests

The authors declare no competing interests.
