## [Peer Review File · Nature Communications]

REVIEWER COMMENTS

Reviewer #1 (Remarks to the Author):

The paper by Zheng et al. describes airborne and ground-based observations in the mid-latitude of the Atlantic Ocean and illustrates the source of new particles in the remote marine boundary layer. Previous observation of the marine new particle formation (NPF) has been attributed to entrainment of new particles from the lower free troposphere into the marine boundary layer. However, aerosol size distribution data along with other auxiliary measurements in this study demonstrate that the NPF events were initiated in the upper, decoupled layer of the marine boundary layer and in the broken cloud fields after the passage of a cold front, when total aerosol surface area was reduced and temperatures were relatively low (less than 5 °C). The authors estimate that these types of NPF events contribute to 50-70% of total CCN and therefore are a large source of CCN in the remote marine boundary layer. The results are very original and are significant for the atmospheric science community, improving our understanding of aerosol-cloud interactions in the marine environments. The paper is written very clearly and extensive supporting information and data from 2 major airborne studies and a long-term ground-based measurements are provided. The one aspect that is not discussed is the data on the nucleation chemical precursors. Adding such evidence (if measured) will strengthen the conclusions even more. I accept the paper for publication after the following minor comments are addressed:

1. L 126: the unit of aerosol surface area should be corrected to $\sim 10 \mu\text{m}^2 \text{cm}^{-3}$.
2. L138-140: Availability of the precursors should be included as one of the required conditions for NPF.
3. L152-154: it is hypothesized that DMS is transported to the upper, decoupled marine layer and its oxidation leads to sulfuric acid and methanesulfonic acid formation and subsequent nucleation. In a recent study by Veres et al (PNAS 2020) a new oxidation species of DMS, hydroxymethyl thioformate was also measured in the marine atmosphere and linked to NPF. Were any of these species measured either on the G1 or the C130? The reference to Veres et al. observation should be added. Adding the vertical profiles of at least DMS would also be valuable.
4. L210: "need close attention...."
5. L294-295: what is the size cut of this stand-alone CPC?
6. L471-472: It seems concentrations of condensable species were available and used for growth rate calculations. Related to my comment #3 above, I suggest including their vertical profiles at least for 1-2 NPF events.

Reviewer #2 (Remarks to the Author):

are used to argue that new particle formation (NPF) is occurring the upper MBL. Observations of the total size distributions and meteorological conditions when small particles were observed are used to suggest a mechanism for this MBL NPF whereby conditions of cloud temperature, high actinic flux and low concentration of pre-existing aerosol coincide following cold fronts, enabling new particles to form from condensable vapors lofted from the ocean surface to the upper decoupled layer by shallow convective clouds. It is claimed that these conditions occur fairly frequently and cover large geographic areas, rendering any cloud condensation nuclei (CCN) resulting from this NPF of global importance.

Observations of particle growth events on Graciosa island are presented as evidence that these small particles formed in the upper MBL grow to sizes where they can act as CCN. Observed and simulated growth rates, and calculated entrainment rates are given as evidence that particles observed growing on Graciosa are formed in the upper MBL, that this growth is widespread over open oceans rather than specific to the Island microcosm and that growth persists to CCN sizes.

NPF has hitherto been thought of as infrequent and sporadic in the MBL, and that small particles in

the MBL were previously thought to originate from the free troposphere (FT). Therefore, this work represents a new understanding of MBL aerosol and remote marine sources of CCN. Because of the large contribution to CCN, these claims are of interest to many in the fields of climate modeling and aerosol-cloud-climate interactions.

Much of the work presented to show that NPF is occurring in the upper MBL under the conditions described above is very convincing. I believe some clarification of the methods chosen to identify NPF from airborne data is needed. My questions regarding these methods are detailed in the following bullet points.

- The authors rely heavily on the constructed variable $N_{>3\text{ nm}}/N_{>10\text{ nm}}$ to indicate NPF, and choose a threshold value of 1.2 to indicate NPF. While this variable makes sense – NPF produces a lot of small particles, some of these grow while others are lost to coagulation, so the size distribution has larger number concentrations at smaller sizes – we also have to consider that larger particles have longer lifetimes. Therefore, we can imagine events where strong nucleation are both occurring, perhaps on top of a background of previous nucleation and growth, such that $N_{>3\text{ nm}}/N_{>10\text{ nm}}$ is closer to 1, but NPF is still occurring. Similarly, weak NPF with weak growth could produce values for $N_{>3\text{ nm}}/N_{>10\text{ nm}}$ above the 1.2 threshold although the event is not going to produce many particles. Therefore, the major conclusions of this paper would be strengthened if the authors could provide more justification for a) using this ratio as opposed to e.g. number concentrations of 3-10 nm particles, and b) justify the choice of 1.2 as threshold and/or demonstrate how varying this threshold affects their conclusions. I could imagine using size distribution from simulated or chamber nucleation events, calculating the resulting $N_{>3\text{ nm}}/N_{>10\text{ nm}}$ and using this to show how robust the variable and chosen threshold are for capturing nucleation under a variety of background conditions/nucleation rates/growth rates.
- $N_{3-10\text{ nm}}$, shown in Fig 1 and related supplementary figures, relies on differencing two large numbers – the total concentrations from two CPCs. What is the detection limit and sensitivity here? I have used a technique similar to this in the past to look for evidence of new particle formation and did some analysis on when the differencing method gave a concentration that was statistically significant (detailed in the Methods section of Williamson 2019 under the sub-heading “Identifying new particle formation”). It would be helpful if the authors could detail why an analysis of statistical significance is not needed in this case, or consider applying an analysis of when this concentration is statistically significant.
- It is technically challenging to maintain consistent flow rates in many CPCs over changes in altitude – it would therefore be helpful if the authors could explain in the methods section how they ensured consistent concentration measurements over the range of altitudes used in the airborne data.
- Is the relationship between the size cut-offs of the two CPCs constant over the range of inlet pressure, concentrations and expected compositions of the aerosols measured? Do the two CPCs both use the same working fluid (i.e. is there any risk of a different response from the CPCs to different aerosol chemical compositions?)
- The figures showing profiles of mean Stot, $N_{3/10}$ and other variables (e.g. Fig 1b) as a function of altitude – it is necessary to see the variability at each altitude level (e.g. error bars for standard deviations or interquartile ranges). Without knowing this variability, for example, the claim that small particles are being formed in-situ in the MBL rather than entrained from the FT because no 3-10 nm particles being observed just above the MBL (line 87), is left a little unsubstantiated.
- In Supplementary Figure 6 D, why is $N_{3-10\text{ nm}}$ not shown as is it for the other examples?

The argument that these particles grow to CCN sizes appears weaker than the rest of the paper, I will detail why I think this in bullet points below, but first would like to stress that, even without a link to CCN sizes, the first part of the paper, widespread NPF in the upper MBL is novel and of much interest to our field.

- On line 37 the authors claim that nucleation mode particles observed during growth events were

not from local sources on Graciosa island. The evidence for this was not clear to me, can the explain further? Have the authors ruled out (e.g. from back-trajectories) any possibility of the particles having nucleated from something like biogenic vapors or anthropogenic activity on any of the other Azores islands?

- The slow growth rate from 1 - ~10 nm referred to on line 378 seem to be the crucial argument for why the growth events seen on the ground site are from particles nucleated over the open ocean, and therefore critical for claiming that these particles form CCN. In supplementary table 1 the growth rate for each observed event is given – how is that calculated? And what size range does it apply to? If the particles are only observed on the island above ~10 nm, how can we infer that these growth rates apply to particles < 10 nm? I believe there may also be a missing word in this sentence which many need to be corrected.
- On line 381 the continuous growth of nucleation mode particles is claimed as evidence for this being a regional phenomenon. Is this because there is evidence of a supply of small particles over a long period of time? This point needs some clarification.
- The raman lidar profiles on event days are said to show “reduced aerosol extinction in the upper decoupled layer” (line 397) – supplementary table 1 shows only the data for event days. Statistics from non-event days should be provided to back up this claim. It would be helpful to also provide easy summary statistics from events days to make the comparison directly.
- Supplementary figure 8b shows a growth event and is cited as evidence of newly formed particles growing to CCN sizes. However, the particles here appear not to grow larger than ~ 40 nm, which is generally considered too small to act as a CCN. Can the authors explain their reasoning in assuming these particles do grow to CCN sizes before being lost to sinks? They claim in line 452 that anomalously strong updrafts could remove enough particles around 50 nm in diameter that they do not show up in the growth bananas. Can the authors explain why this is the case here, but other ground site observations of new particle formation and growth see particles well beyond 50 nm? Furthermore, the authors show the Hoppel minimum here to be at 80 nm (line 458), which also suggests that we should still see more 50 nm particles than are present in the size distribution. I’m struggling to follow this part of the argument and would appreciate some further explanation.
- The simulated growth mentioned in line 472 – how does this compared with the measured growth? Could this be shown in a figure? I’m also unclear about why in the simulations particles grow to 80 nm in clear air, but in the observed size distributions they do no grow this large. To me that would indicate a difference between the simulations and observations that brings into question the quote 50% and 70% of CCN (line 487).
- Some more stats are needed on the CCN contribution from NPF e.g. the range or standard deviation to give some idea of how much this varies.

There are some points that need clarification regarding how widespread and important this method of CCN production is:

- The paper claims that because favorable conditions for NPF are widespread, and growth to CCN sizes are observed from Graciosa island, that production of CCN from this NPF is also widespread. As the paper stands, it is not clear to me that the possibility that condensable vapors associated with the island (e.g. coastal, anthropogenic, terrestrial-biogenic) contribute substantially to the growth, has been sufficiently ruled out for this claim to be made. Can the authors explain this further or supply more evidence?
- Line 38: ref 2 is cited to say that CCN concentrations are responsible for a significant fraction of variability in MBL radiative effects, but I believe this reference actually says that CCN effects on albedo variance is small when compared with factors such as natural meteorological variability. While the authors correctly cite ref. 3, which supports their argument that CCN are important here, they need to address the fact that there are conflicting arguments in the literature on the importance of CCN variation for albedo modification in the MBL.
- The contribution of newly formed particles to CCN is stated as 50-70% on line 189. My understanding from the text is that this is instantaneous i.e. at the end of the NPF + growth event and local. To understand the importance of MBL CCN we need to know not only the instantaneous

contribution to CCN locally, but also some estimation of the contribution globally over longer timescales. It would be very helpful if the authors could tie this together with what they mention about the conditions for this phenomenon being both widespread and frequent to estimate also the average contribution annually and globally (or as a function of the total annual MBL CCN). While there will be large uncertainties in this estimate, it would give context for the importance of this new mechanism. I am also struggling to understand where the 50% and 70% quoted here are calculated. To me it is not clear from fig 4.

- Line 116 states that upper MBL NPF occurs over a wide area. This could be better supported if a map of all profiles measured where NPF was and wasn't observed along with cloud height, Stot etc.

In the main data, the total surface area is used as a proxy for the coagulation sinks. I have a few questions about this:

- Why did the authors choose to use Stot instead of calculating the coagulation sink for e.g. 3 nm particles, from the size distribution? If they wish to use this proxy instead of a calculated sink, a supplementary figure showing its close correlation to the coagulation sink for relevant sizes would make this choice more justified.

- In the methods section (Line 284) the authors mention that Stot is calculated for particles larger than 10, sometimes above 100 nm, and sometimes using size distributions from the LAS instrument instead of the FIMS. While larger particles generally dominate the total coagulation sink, during nucleation events in clean conditions, the sink to smaller particles can become important. Some characterization of how much this could influence the results (e.g. using simulated nucleation events under conditions similar to the observations) would remove questions about biases in the data due to these unavoidable differences in instrumentation. A 3 – 10 nm size bin, using the CPC data could even be included in the calculated Stot or coagulation sink this analysis shows the contribution to the sink from this size range to be important.

Finally, a few smaller unrelated questions:

- The authors mention that only clear-sky data are used to avoid inlet artifacts. How are in-cloud and out-of-cloud time periods identified. A strong explanation this classification seems essential since we are seeing large numbers of small particles in the vicinity of clouds and the community will be keen to rule out any possibility of these being artifacts from measuring inside clouds.

- I believe a small number technical correction is needed in line 126 - unit should be μm^2 instead of μm^{-2}

I believe this paper, once the issues mentioned above are addressed, will influence thinking in modeling marine clouds, understanding aerosol sources and the importance of new particle formation. It challenges the long-held view that sinks are too high in the MBL for NPF to be an important contributor to CCN here.

References

Williamson, C.J., Kupc, A., Axisa, D. et al. A large source of cloud condensation nuclei from new particle formation in the tropics. *Nature* 574, 399–403 (2019). <https://doi.org/10.1038/s41586-019-1638-9>

Signed: Christina J Williamson

Reviewer #3 (Remarks to the Author):

This MS reports on observations of new particle formation in North Atlantic marine boundary layer (MBL), more specifically in the upper decoupled layer, preceded by the passage of a cold front. Previously it has been thought that secondary new particles observed in MBL are formed in free troposphere (above upper MBL layer) and mixed with MBL. Results that NPF occur in MBL also and not only in FT is important for proper description of NPF in models.

Explanation of observed NPF which is not caught by the models is much lower aerosol surface area in the upper MBL than assumed in the models. This work certainly should help improving models and in that sense deserves to be published. However, I do not know, if the result that NPF occurs in the atmospheric layer (upper decoupled MBL) where it was not believed to be frequent rather than (just) in FT above the upper decoupled MBL is remarkable in the eyes of wider audience or would these results suit better for a more specific journal.

While the key results are based on observations made during few tens of Gulfstream flights, scaling to wider geographical areas seem justified in the light of year-long measurements in southern Atlantic previous research cruises in the Pacific and Indian Oceans where new particles were frequently observed in similar meteorological situations (passage of cold front). Thus the conclusions are scientifically sound.

The mechanism of NPF is speculative. Concurrent vapour measurements would have added much to the study. Authors suggest DMS oxidation products (sulphuric and methane sulphonic acid) to be responsible on particle formation which is likely a correct assumption. However, without trace gas / vapour data, the mechanism, including the role of ammonia etc. bases and air ionisation remain speculative.

We thank the referees for their thoughtful and constructive comments. Please find below detailed responses to each comment or question, including notations of improvements to the manuscript. Referee comments are in blue fonts. Changes to the manuscript are shown in indented text. The line numbers used here refer to those in the manuscript file with track changes shown.

5

Reviewer #1 (Remarks to the Author):

Comment:

10 The paper by Zheng et al. describes airborne and ground-based observations in the mid-latitude of the Atlantic Ocean and illustrates the source of new particles in the remote marine boundary layer. Previous observation of the marine new particle formation (NPF) has been attributed to entrainment of new particles from the lower free troposphere into the marine boundary layer. However, aerosol size distribution data along with other auxiliary measurements in this study demonstrate that the NPF events were initiated in the upper, decoupled layer of the marine boundary layer and in the broken cloud fields after the passage of a cold front, when total aerosol surface area was reduced and temperatures were relatively low (less than 5 °C). The authors estimate that these types of NPF events contribute to 50-70% of total CCN and therefore are a large source of CCN in the remote marine boundary layer. The results are very original and are significant for the atmospheric science community, improving our understanding of aerosol-cloud interactions in the marine environments. The paper is written very clearly and extensive supporting information and data from 2 major airborne studies and a long-term ground-based measurements are provided. The one aspect that is not discussed is the data on the nucleation chemical precursors. Adding such evidence (if measured) will strengthen the conclusions even more. I accept the paper for publication after the following minor comments are addressed:

15

20

Response:

25 We thank the reviewer for the positive and constructive comments, which have led to a substantial improvement of the manuscript. Below please find our point-by-point response.

Comment:

1. L 126: the unit of aerosol surface area should be corrected to $\sim 10 \mu\text{m}^2 \text{ cm}^{-3}$.

30

Response:

The unit has been corrected and we have checked all units throughout the manuscript.

Comment:

35 2. L138-140: Availability of the precursors should be included as one of the required conditions for NPF.

Response:

We agree and have revised the manuscript accordingly. It now reads (main text line 144):

40 “The above results and the vertical profiles of DMS mixing ratio (e.g., Fig. 1B, Supplementary Fig. 7B) indicate that NPF in the MBL is due to a combination of low S_{tot} , cold ambient temperature, availability of the precursors, and photo-oxidation within the clear regions of open-cell convection or a scattered cloud field.”

Comment:

45 3. L152-154: it is hypothesized that DMS is transported to the upper, decoupled marine layer and its oxidation leads to sulfuric acid and methanesulfonic acid formation and subsequent nucleation. In a recent study by Veres et al (PNAS 2020) a new oxidation species of DMS, hydroxymethyl thioformate was also measured in the marine atmosphere and linked to NPF. Were any of these species measured either on the G1 or the C130? The reference to Veres et al. observation should be added. Adding the vertical profiles of at least DMS would also be
50 valuable.

Response:

The vertical profiles of the DMS mixing ratio during the new particle formation events have been added in Fig. 1 and relevant supplementary figures. In general, the vertical profiles of DMS resemble those of water vapor, showing elevated mixing ratios in both surface mixed layer and
55 upper decoupled layer compared to that in the free troposphere. This is expected as DMS originates from the ocean surface. The elevated DMS mixing ratio in the upper decoupled layer of the MBL suggests that the oxidation products of DMS including H_2SO_4 likely contribute to the new particle formation observed. The gas-phase DMS mixing ratio was measured by a PTR-ToF-MS during the NAAMES flights. During ACE-ENA, the mixing ratio of DMS was
60 measured by an Ionicon quadrupole PTR-MS onboard the G-1. Because the quadrupole PTR-MS cycled through a number of pre-selected m/z values, the duty cycle of the PTR-MS for individual species is lower than that in the PTR-ToF-MS, leading to reduced signal to noise ratio. In addition, given the unit mass resolution of the quadrupole PTR-MS, isobaric interference may also lead to a positive bias in the measured DMS mixing ratio. Here we only use the vertical
65 profile of the ion signal at m/z 63 measured onboard the G-1 to qualitatively infer the relative abundance of DMS in the upper part of MBL. The conclusion of this study is not based on the

absolute values of the DMS mixing ratio, and therefore is not affected. The above description and discussion have been included in the Results and discussion (line 80):

70 “The ion signal at m/z 63 measured by a Proton-Transfer-Reaction Mass Spectrometer (PTR-MS) suggests that vertical profile of dimethyl sulfide (DMS) is similar to that of water vapor, showing elevated mixing ratio in both surface mixed layer and upper decoupled layer compared to that in the free troposphere (Fig. 1B). This is expected as DMS originates from the ocean surface.”

and Methods section as (line 267):

75 “An Ionicon quadrupole high-sensitivity Proton-Transfer-Reaction Mass Spectrometer (abbreviated as PTR-MS hereinafter) was used to measure the mixing ratios of selected gas-phase VOC including DMS. Because the quadrupole PTR-MS cycled through a number of pre-selected m/z values, the duty cycle for each m/z is limited, leading to a reduced signal to noise ratio. In addition, given the unit mass resolution of the quadrupole PTR-MS, isobaric
80 interference may also lead to a positive bias in the measured DMS mixing ratio. Here we only use the vertical profile of the ion signal at m/z 63 measured onboard the G-1 to qualitatively infer the relative abundance of DMS in the upper part of MBL. The conclusion of this study does not depend on the absolute values of the DMS mixing ratio.”

Unfortunately, hydroperoxymethyl thioformate was not measured onboard G1 during ACE-ENA
85 or C130 during NAAMES. We do agree that hydroperoxymethyl thioformate¹ may contribute to the new particle formation observed. We have discussed this mechanism and added the reference to the manuscript (line 162) as:

90 “A recent study discovered a new oxidation product of DMS, hydroperoxymethyl thioformate, which may also contribute to the NPF and particle growth in marine environment³¹.”

Comment:

4. L210: “need close attention....”

Response:

95 Thanks for the correction. We’ve revised the manuscript accordingly.

Comment:

5. L294-295: what is the size cut of this stand-alone CPC?

Response:

100 The cut-off size of this standalone CPC is 10 nm. The sentence has been modified to (line 321):

“The aerosol number concentration (CN) was measured concurrently by a standalone CPC (Model 3772, TSI Inc.) with a 50% cut-off size of 10 nm.”

Comment:

105 6. L471-472: It seems concentrations of condensable species were available and used for growth rate calculations. Related to my comment #3 above, I suggest including their vertical profiles at least for 1-2 NPF events.

Response:

110 Sorry for the confusion. Unfortunately, we don't have direct measurements of the condensing species at the ENA site. In L471-472, we meant that we derived the growth rate (GR) between ~35 nm and 80 nm based on the observed growth rate below ~35 nm, as GR is a function of particle diameter (D_p) at the same vapor pressure of nonvolatile condensing species. The growth rate, i.e. rate of diameter change of a particle due to condensation, can be calculated as²:

$$GR(D_p) = \frac{dD_p}{dt} = \frac{4D_i M_i}{RT \rho_p} \cdot \frac{f(Kn, \alpha)}{D_p} \cdot (p_i - p_{eq,i}) \quad (R1a)$$

115 where

$$f(Kn, \alpha) = \frac{0.75\alpha(1 + Kn)}{Kn^2 + (1 + 0.283\alpha)Kn + 0.75\alpha} \quad (R1b)$$

120 where D_i is the mass diffusivity of condensate i in air, M_i is its molecular weight, R is the gas constant, T is temperature in K, ρ_p is the particle density. p_i and $p_{eq,i}$ are the vapor pressures of condensate i in the bulk gas-phase and at the aerosol surface, respectively. For a nonvolatile condensate like H_2SO_4 , $p_{eq,i}$ is zero³. Kn is the Knudsen number defined as $2\lambda_{mfp}/D_p$, where λ_{mfp} is the mean free path. The term $f(Kn, \alpha)$ is the correction due to non-continuum effects (depending on Kn) and imperfect surface accommodation (depending on mass accommodation coefficient α), and is estimated by the Fuchs-Sutugin approach².

125 The above equation shows that at the same condensate vapor pressure, the size dependence of GR originates from the term $f(Kn, \alpha)D_p^{-1}$. When D_p is below 35 nm, this dependence is negligible and GR can be estimated from a linear fitting of the time series of $D_{p,n}$. In the size range of 35 to

80 nm, GR is derived assuming the same condensate vapor pressure, and with the size dependence taken into account using the term $f(Kn, \alpha)D_p^{-1}$.

We've included this information into the revised manuscript as (line 627):

130 “For the simulation of condensational growth, the particle growth rate (GR) is derived from the size distribution measurement using the following approach. The GR is a function of particle diameter D_p (ref.²):

$$GR(D_p) = \frac{dD_p}{dt} = \frac{4D_i M_i}{RT \rho_p} \cdot \frac{f(Kn, \alpha)}{D_p} \cdot (p_i - p_{eq,i}) \quad (8a)$$

where

135
$$f(Kn, \alpha) = \frac{0.75\alpha(1 + Kn)}{Kn^2 + (1 + 0.283\alpha)Kn + 0.75\alpha} \quad (8b)$$

where D_i is the mass diffusivity of condensate i in air, M_i is its molecular weight, R is the gas constant, T is temperature in K, ρ_p is the particle density. p_i and $p_{eq,i}$ are the vapor pressures of condensate i in the bulk gas-phase and at the aerosol surface, respectively. For a nonvolatile condensate like H_2SO_4 , $p_{eq,i}$ is zero³. Kn is the Knudsen number defined as $2\lambda_{mfp}/D_p$, where λ_{mfp} is the mean free path. The term $f(Kn, \alpha)$ is the correction due to non-continuum effects (depending on Kn) and imperfect surface accommodation (depending on mass accommodation coefficient α), and is estimated by the Fuchs-Sutugin approach².

140

Eq. 8a shows that at the same condensate vapor pressure, the size dependence of GR originates from the term $f(Kn, \alpha)D_p^{-1}$. As this size dependence is negligible below 35 nm, the initial growth rate is first derived from a linear fit of the nucleation mode diameter ($D_{p,n}$) with time, until $D_{p,n}$ reaches 35 nm or in some cases, the growth is interrupted by air mass change. The rate of subsequent particle growth, $GR(D_p)$, from ~ 35 nm to 80 nm is then calculated assuming the same vapor pressure of nonvolatile condensing species (e.g., sulfuric acid or condensable organics) with the size dependence of condensational growth (i.e., $f(Kn, \alpha)D_p^{-1}$) taken into account.”

145

150

Reviewer #2 (Remarks to the Author):

General comment:

155 are used to argue that new particle formation (NPF) is occurring the upper MBL. Observations of the total size distributions and meteorological conditions when small particles were observed are used to suggest a mechanism for this MBL NPF whereby conditions of cloud temperature, high actinic flux and low concentration of pre-existing aerosol coincide following cold fronts, enabling new particles to form from condensable vapors lofted from the ocean surface to the upper decoupled layer by shallow convective clouds. It is claimed that these conditions occur
160 fairly frequently and cover large geographic areas, rendering any cloud condensation nuclei (CCN) resulting from this NPF of global importance.

Observations of particle growth events on Graciosa island are presented as evidence that these small particles formed in the upper MBL grow to sizes where they can act as CCN. Observed and simulated growth rates, and calculated entrainment rates are given as evidence that particles
165 observed growing on Graciosa are formed in the upper MBL, that this growth is widespread over open oceans rather than specific to the Island microcosm and that growth persists to CCN sizes. NPF has hitherto been thought of as infrequent and sporadic in the MBL, and that small particles in the MBL were previously thought to originate from the free troposphere (FT). Therefore, this work represents a new understanding of MBL aerosol and remote marine sources of CCN.
170 Because of the large contribution to CCN, these claims are of interest to many in the fields of climate modeling and aerosol-cloud-climate interactions.

Response:

We thank Dr. Williamson for her positive and constructive comments, which led to a substantially improved manuscript. Below please find our point-by-point response.

175

Comments:

Much of the work presented to show that NPF is occurring in the upper MBL under the conditions described above is very convincing. I believe some clarification of the methods chosen to identify NPF from airborne data is needed. My questions regarding these methods are
180 detailed in the following bullet points.

- The authors rely heavily on the constructed variable $N_{>3\text{ nm}}/N_{>10\text{ nm}}$ to indicate NPF, and choose a threshold value of 1.2 to indicate NPF. While this variable makes sense – NPF produces a lot of small particles, some of these grow while others are lost to coagulation, so the size
185 distribution has larger number concentrations at smaller sizes – we also have to consider that

larger particles have longer lifetimes. Therefore, we can imagine events where strong nucleation are both occurring, perhaps on top of a background of previous nucleation and growth, such that $N_{>3\text{ nm}}/N_{>10\text{ nm}}$ is closer to 1, but NPF is still occurring. Similarly, weak NPF with weak growth could produce values for $N_{>3\text{ nm}}/N_{>10\text{ nm}}$ above the 1.2 threshold although the event is not going to produce many particles. Therefore, the major conclusions of this paper would be strengthened if the authors could provide more justification for a) using this ratio as opposed to e.g. number concentrations of 3-10 nm particles, and b) justify the choice of 1.2 nm as threshold and/or demonstrate how varying this threshold affects their conclusions. I could imagine using size distribution from simulated or chamber nucleation events, calculating the resulting $N_{>3\text{ nm}}/N_{>10\text{ nm}}$ and using this to show how robust the variable and chosen threshold are for capturing nucleation under a variety of background conditions/nucleation rates/growth rates.

Response:

We thank Dr. Williamson for this thoughtful and constructive comment. $N_{>3\text{nm}}/N_{>10\text{nm}}$ and $N_{>3\text{nm}}-N_{>10\text{nm}}$ have been used in different studies to identify NPF⁴⁻⁷. One key reason to use $N_{>3\text{nm}}/N_{>10\text{nm}}$ here is because the main focus of this study is to show that NPF occurs in the upper part of the MBL instead of the FT as previously thought. As pointed out by Dr. Williamson, for scenarios when the concentration of existing particles larger than 10 nm is high, there could be a substantial number of particles between 3 and 10 nm even when the ratio $N_{>3\text{nm}}/N_{>10\text{nm}}$ is close to 1. However, it is more challenging to exclude the possibility that these small particles (i.e., the particles between 3 and 10 nm) were initially formed at other altitudes before being transported to where they were observed. For example, if a substantial number of 3-10 nm particles were observed in the upper MBL, it is possible (albeit unlikely) that these particles were originally formed elsewhere (e.g., in lower free troposphere) before being transported to the upper MBL. To the first order, $N_{>3\text{nm}}/N_{>10\text{nm}}$ describes the shape of aerosol size distribution. An elevated $N_{>3\text{nm}}/N_{>10\text{nm}}$ indicates a large fraction of 3-10 nm particles, more clearly indicating a relatively early stage (i.e., recent) of new particle formation. Therefore, using the vertical profile of $N_{>3\text{nm}}/N_{>10\text{nm}}$ allows us to better examine and identify the altitudes where NPF occurs.

We completely agree that, while a threshold value of 1.2 makes sense, it is somewhat arbitrary. We have modified the threshold by employing the same approach described in Williamson et al. (2019). The approach is described as follows. We group the 1-s measurements into 10 s intervals, which correspond to a spatial scale of 1 km. For each 10-second interval, the ratio of average $N_{>3\text{nm}}$ to average $N_{>10\text{nm}}$ (R) and the standard deviation of the ratio (σ_R) are derived. A NPF event is identified when the ratio $N_{>3\text{nm}}/N_{>10\text{nm}}$ is statistically significant:

$$N_{>3\text{nm}}/N_{>10\text{nm}} > 1 + 3 \sigma_{\text{R}} \quad (\text{R2})$$

σ_{R} over a sampling period of Δt is written as:

$$\sigma_{\text{R}} = \sigma\left(\frac{N_{>3\text{nm},\Delta t}}{N_{>10\text{nm},\Delta t}}\right) \quad (\text{R3})$$

220 where $N_{>3\text{nm},\Delta t}$ and $N_{>10\text{nm},\Delta t}$ are the average $N_{>3\text{nm}}$ and $N_{>10\text{nm}}$ over the time interval of Δt . Based on uncertainty propagation, σ_{R} is given by:

$$\begin{aligned} \sigma_{\text{R}} &= R \sqrt{\left(\frac{\sigma(N_{>3\text{nm},\Delta t})}{N_{>3\text{nm},\Delta t}}\right)^2 + \left(\frac{\sigma(N_{>10\text{nm},\Delta t})}{N_{>10\text{nm},\Delta t}}\right)^2} \\ &= \frac{N_{>3\text{nm},\Delta t}}{N_{>10\text{nm},\Delta t}} \sqrt{\left(\frac{\sigma(N_{>3\text{nm},\Delta t})}{N_{>3\text{nm},\Delta t}}\right)^2 + \left(\frac{\sigma(N_{>10\text{nm},\Delta t})}{N_{>10\text{nm},\Delta t}}\right)^2} \end{aligned} \quad (\text{R4})$$

For CPC measurements, the particle concentration ($N_{\Delta t}$) is based on $N_{\Delta t} = C_{\Delta t} / (Q\Delta t)$, where $C_{\Delta t}$ is the number of particle counts measured over the time period of Δt and Q is the aerosol sample flow rate of the CPC. We assume that the particle counts are described by Poisson statistics, such that $\sigma(C_{\Delta t}) = \sqrt{C_{\Delta t}}$ and $\sigma(N_{\Delta t}) = \sigma(C_{\Delta t}) / (Q\Delta t) = \sqrt{C_{\Delta t}} / (Q\Delta t)$. Therefore, we can derive that

$$\sigma^2(N_{\Delta t}) = \frac{C_{\Delta t}}{(Q\Delta t)^2} = \frac{N_{\Delta t}}{Q\Delta t} \quad (\text{R5})$$

Combing Eq. (R4) and (R5), we have:

$$\sigma_{\text{R}} = \frac{N_{>3\text{nm},\Delta t}}{N_{>10\text{nm},\Delta t}} \sqrt{\frac{1}{C_{>3\text{nm},\Delta t}} + \frac{1}{C_{>10\text{nm},\Delta t}}} \quad (\text{R6})$$

A NPF event is identified when $N_{>3\text{nm}}/N_{>10\text{nm}}$ over a 10-s interval is statistically above 1 (i.e., $N_{>3\text{nm}}/N_{>10\text{nm}} > 1 + 3 \sigma_{\text{R}}$). This approach identifies the same new particle formation events as the previous one (i.e., using a fixed threshold of 1.2). The discussion below has been included in the manuscript (line 389):

235 “To identify NPF events, we first group the 1-s measurements into 10 s intervals, which correspond to a spatial scale of 1 km. For each 10-second interval, the ratio of average $N_{>3\text{nm}}$ to average $N_{>10\text{nm}}$ (R) and the corresponding uncertainty of the ratio (σ_R) are derived. A NPF event is identified when the ratio $N_{>3\text{nm}}/N_{>10\text{nm}}$ is statistically significant:

$$N_{>3\text{nm}}/N_{>10\text{nm}} > 1 + 3 \sigma_R \quad (1)$$

σ_R over a sampling period of Δt is written as:

$$\sigma_R = \sigma\left(\frac{N_{>3\text{nm},\Delta t}}{N_{>10\text{nm},\Delta t}}\right) \quad (2)$$

240 where $N_{>3\text{nm},\Delta t}$ and $N_{>10\text{nm},\Delta t}$ are average $N_{>3\text{nm}}$ and $N_{>10\text{nm}}$ over the time interval of Δt . Based on uncertainty propagation, σ_R is given by:

$$\sigma_R = \frac{N_{>3\text{nm},\Delta t}}{N_{>10\text{nm},\Delta t}} \sqrt{\left(\frac{\sigma(N_{>3\text{nm},\Delta t})}{N_{>3\text{nm},\Delta t}}\right)^2 + \left(\frac{\sigma(N_{>10\text{nm},\Delta t})}{N_{>10\text{nm},\Delta t}}\right)^2} \quad (3)$$

245 For CPC measurements, the particle concentration ($N_{\Delta t}$) is based on $N_{\Delta t} = C_{\Delta t}/(Q\Delta t)$, where $C_{\Delta t}$ is the number of particle counts measured over the time period of Δt and Q is the aerosol sample flow rate of the CPC. We assume that the particle counts are described by Poisson statistics, such that $\sigma(C_{\Delta t}) = \sqrt{C_{\Delta t}}$ and $\sigma(N_{\Delta t}) = \sigma(C_{\Delta t})/(Q\Delta t) = \sqrt{C_{\Delta t}}/(Q\Delta t)$. Therefore, we can derive that

$$\sigma^2(N_{\Delta t}) = \frac{C_{\Delta t}}{(Q\Delta t)^2} = \frac{N_{\Delta t}}{Q\Delta t} \quad (4)$$

Combining Eqs. (3) and (4), we have:

$$\sigma_R = \frac{N_{>3\text{nm},\Delta t}}{N_{>10\text{nm},\Delta t}} \sqrt{\frac{1}{C_{>3\text{nm},\Delta t}} + \frac{1}{C_{>10\text{nm},\Delta t}}} \quad (5)$$

250 A NPF event is identified when $N_{>3\text{nm}}/N_{>10\text{nm}}$ over a 10-s interval is statistically above 1 (i.e., $N_{>3\text{nm}}/N_{>10\text{nm}} > 1 + 3 \sigma_R$). The analysis is focused on periods with BC mass concentration below 2 ng m^{-3} to exclude the occasional impact of local pollution from the islands.”

Comment:

255 • N3-10 nm, shown in Fig 1 and related supplementary figures, relies on differencing two large numbers – the total concentrations from two CPCs. What is the detection limit and sensitivity here? I have used a technique similar to this in the past to look for evidence of new particle

260 formation and did some analysis on when the differencing method gave a concentration that was statistically significant (detailed in the Methods section of Williamson 2019 under the sub-heading “Identifying new particle formation”). It would be helpful if the authors could detail why an analysis of statistical significance is not needed in this case, or consider applying an analysis of when this concentration is statistically significant.

Response:

265 We thank Dr. Williamson for this suggestion. We have followed the approach detailed in Williamson et al. (2019)⁸, and derived the uncertainty of $N_{>3\text{nm}}/N_{>10\text{nm}}$ (σ_R) for each altitude bin. Elevated $N_{>3\text{nm}}/N_{>10\text{nm}}$ that are statistically significant (i.e., $>1 + 3 \sigma_R$) are now distinguished using a different marker (i.e., filled square) in Fig. 1 and related supplementary figures. The derivation of the uncertainty of $N_{>3\text{nm}}/N_{>10\text{nm}}$ (σ_R) is detailed in the response to the previous comment and the Methods section.

270

Comment:

275 • It is technically challenging to maintain consistent flow rates in many CPCs over changes in altitude – it would therefore be helpful if the authors could explain in the methods section how they ensured consistent concentration measurements over the range of altitudes used in the airborne data.

Response:

280 We thank Dr Williamson for bringing up this important point regarding data quality control/assessment for the deployed CPCs: a CPC 3772 ($D_p > 10$ nm), and a CPC 3025A ($D_p > 3$ nm). Our confidence in the reported CPC concentrations and the concentration ratio is based on several considerations:

1. CPC measurements relevant to NPF occurred at altitudes below ~ 2000 m, limiting/constraining the variability in sampled altitude/atmospheric pressure (i.e., pressure between 80 kPa and 101 kPa) that might impact CPC flow rate stability.
2. The sample flow rate of the CPC 3772 is controlled via a critical orifice and is expected to be constant over the relevant sampling altitude range (i.e., < 2000 m).
3. The sample flow rate of the CPC 3025A is not actively controlled and maintaining flow stability can be a challenge. However, during the periods when newly formed particles were absent (based on the shape of aerosol size distribution), the number concentration ratio of the CPC 3025A to the CPC 3772 (i.e., $N_{>3\text{nm}}/N_{>10\text{nm}}$) does not change appreciably with sampling

290 altitude, indicating flow rate stability in the CPC 3025A over the relevant sampling altitude range.

Related discussion has been added in the Methods section (line 411):

295 “We note that the measured $N_{>3\text{nm}}$ and $N_{>10\text{nm}}$, and therefore the derived ratio $N_{>3\text{nm}}/N_{>10\text{nm}}$ may be impacted by flow instability due to sampling at different altitudes and the variation in the CPC 50% cut-off sizes due to the changes in the altitude (i.e., sampling pressure), particle concentration, and composition. In this study, the NPF in the upper part of MBL were observed below the altitude of approximately 2000 m, corresponding to a pressure range of 80 to 101 kPa. Over this relatively narrow pressure range, the flow instability of the CPCs is negligible, and the variation in cut-off sizes for the CPC 3772 and CPC 3025A is also expected to be very minor. The sample flow rate of the CPC 3772 was controlled via a critical orifice and is expected to be constant over the narrow sampling pressure range. Whereas the sample flow rate of the CPC 3025A is not actively controlled, during the periods when newly formed particles were absent based on the spectral shape of the size distribution measured by the FIMS, $N_{>3\text{nm}}/N_{>10\text{nm}}$ did not change appreciably with sampling altitude, indicating flow-rate stability in the CPC 3025A over the relevant sampling altitude range. Although CPC size cut-offs tend to increase with increasing concentration of the sampled aerosol particles, aerosol number concentrations during this study were less than $\sim 5000 \text{ cm}^{-3}$, substantially below the concentration thresholds above which the impact on cut-off sizes is expected. In addition, as both CPCs use the same working fluid (butanol), we do not expect that the particle composition-dependence of cut-off sizes strongly impacts the interpretation of $N_{>3\text{nm}}/N_{>10\text{nm}}$.”

Comment:

315 • Is the relationship between the size cut-offs of the two CPCs constant over the range of inlet pressure, concentrations and expected compositions of the aerosols measured? Do the two CPCs both use the same working fluid (i.e. is there any risk of a different response from the CPCs to different aerosol chemical compositions?)

Response:

320 Again, an important set of considerations brought up by the reviewer that can impact the interpretation of the reported CPC concentrations. Our confidence in the reported CPC concentration ratio is based on several considerations:

1. The NPF events in the upper part of the MBL were observed below the altitude of ~ 2000 m, corresponding to a pressure range of ~ 80 kPa to 101 kPa. Over this relatively narrow pressure

range, the variation of cut-off sizes for the CPC 3772 and CPC 3025A is very minor. For the
325 CPC 3025A, the cut-off size increases from 2.5 nm at a sampling pressure of 101 kPa to ~2.8 nm
at 80 kPa. There is no notable change of the CPC 3772 cut-off size in the pressure change of 80
to 101 kPa⁹. Therefore, these minor variations should not appreciably impact our interpretation
of what the ratio $N_{>3\text{nm}}/N_{>10\text{nm}}$ means and what it is used for in this study: to assist in identifying
when and where NPF occurs.

330 2. CPC cut-off sizes tend to increase with increasing sampled aerosol number concentration. For
this study, however, sampled aerosol number concentrations were less than $\sim 5000 \text{ cm}^{-3}$,
substantially below the thresholds above which we would expect impacts on the cut-off sizes.

3. Indeed, CPC cut-off sizes vary with particle composition. What gives us confidence in the
derived concentration ratio is that both CPCs use the same working fluid (butanol), so we do not
335 expect that the particle composition-dependence strongly impacts the interpretation of
 $N_{>3\text{nm}}/N_{>10\text{nm}}$.

Related discussion has been added in the Methods section (line 414):

“In this study, the NPF in the upper part of the MBL were observed below the altitude of
approximately 2000 m, corresponding to a pressure range of 80 to 101 kPa. Over this
340 relatively narrow pressure range, the flow instability of the CPCs is negligible, and the
variation in cut-off sizes for the CPC 3772 and CPC 3025A is also expected to be very
minor. The sample flow rate of the CPC 3772 was controlled via a critical orifice and is
expected to be constant over the narrow sampling pressure range. Whereas the sample flow
rate of the CPC 3025A is not actively controlled, during the periods when newly formed
345 particles were absent based on the spectral shape of the size distribution measured by the
FIMS, $N_{>3\text{nm}}/N_{>10\text{nm}}$ does not change appreciably with sampling altitude, indicating flow-rate
stability in the CPC 3025A over the relevant sampling altitude range. Although CPC size
cut-offs tend to increase with increasing concentration of the sampled aerosol particles,
aerosol number concentrations during this study were less than $\sim 5000 \text{ cm}^{-3}$, substantially
350 below the concentration thresholds above which the impact on cut-off sizes is expected. In
addition, as both CPCs use the same working fluid (butanol), we do not expect that the
particle composition-dependence of cut-off sizes strongly impacts the interpretation of
 $N_{>3\text{nm}}/N_{>10\text{nm}}$.”

355 **Comment:**

• The figures showing profiles of mean Stot, N3/N10 and other variables (e.g. Fig 1b) as a
function of altitude – it is necessary to see the variability at each altitude level (e.g. error bars for

standard deviations or interquartile ranges). Without knowing this variability, for example, the claim that small particles are being formed in-situ in the MBL rather than entrained from the FT because no 3-10 nm particles being observed just above the MBL (line 87), is left a little unsubstantiated

360

Response:

We thank Dr. Williamson for this suggestion. The error bars for standard deviations are now added to Figs. 1B, 1C and relevant supplementary figures.

365

Comment:

• In Supplementary Figure 6 D, why is $N_{3-10\text{nm}}$ not shown as is it for the other examples?

Response:

The concentrations of particles between 3 and 10 nm have been added to Supplementary Fig. 6E. As discussed in the manuscript, this is the case with $N_{>3\text{nm}}/N_{>10\text{nm}}$ close to 1 and low concentration of particles between 3 and 10 nm. However, the nucleation mode particles in the upper decoupled layer exhibited a substantially smaller mode diameter than the aerosol immediately above the MBL, indicating that the nucleation mode particles grew from new particles formed in the MBL instead of having been entrained from the FT.

370

375

Comment:

The argument that these particles grow to CCN sizes appears weaker than the rest of the paper, I will detail why I think this in bullet points below, but first would like to stress that, even without a link to CCN sizes, the first part of the paper, widespread NPF in the upper MBL is novel and of much interest to our field.

380

• On line 37 the authors claim that nucleation mode particles observed during growth events were not from local sources on Graciosa island. The evidence for this was not clear to me, can the explain further? Have the authors ruled out (e.g. from back-trajectories) any possibility of the particles having nucleated from something like biogenic vapors or anthropogenic activity on any of the other Azores islands?

385

Response:

Following Dr. Williamson's suggestion, we've clarified and provided additional evidence that the nucleation mode particles observed during the growth events were not from local sources.

First, the continuous growth of nucleation mode particles over periods of several hours or more has been attributed to regional-scale new particle formation events¹⁰ by many earlier studies in

390

both clean¹¹⁻¹⁴ and polluted environments^{15,16}. For most of the events observed at the ENA site, continuous growth of nucleation mode particles was observed despite substantial shift in wind direction, suggesting regional scale events. As these regional-type events occur over relatively large spatial scales, they are unlikely due to local point sources from the Azores islands. The relatively large scale of these new particle formation events is supported by the aircraft measurements (Fig. 1D and Supplementary Fig. 7C).

Second, following Dr. Williamson's suggestion, we've calculated 3-day back-trajectories of air masses arriving at 100 m above the ground level at the ENA site, for each hour during the observed growth events. We found that for over 50% of the events, the trajectories didn't intersect with any of the Azores islands (Fig. R1a). For only 12% of the events, more than 50% of the hourly air mass trajectories during the event passed over one or more of the Azores islands (Fig. R1a). However, the particle growth rate exhibits no dependence on the fraction of hourly trajectories that passed over the Azores islands during the event (Fig. R1b), indicating that the observed nucleation mode particles and particle growth are unlikely due to the biogenic emissions or anthropogenic activities on surrounding Azores islands.

In addition, the observed events are unlikely due to the local sources on Graciosa Island (a small island where the ENA site is located), as discussed in the SI. The above lines of evidence indicate that the observed growths of nucleation mode particles are regional events over the open ocean.

410 **Figure R1. Particle growth rate is independent of the fraction of the air masses passing through the Azores islands during the growth events.** (a) Numbers of events with 0%, 0-50%, and >50% of the hourly trajectories passing through at least one of the Azores islands, and (b) statistics of particle growth rate for the different fractions. The box-whisker plot is drawn for 10-, 25-, 50-, 75-, and 90-percentiles.

We've modified the description in the main text as (line 478):

“The nucleation mode particles during these growth events were not from local sources. Instead, the growth events are regional phenomena based on the following lines of evidence. First, the continuous growth of nucleation mode particles over periods of several hours or more has been attributed to regional-scale new particle formation events¹⁰ by many earlier studies in both clean¹¹⁻¹⁴ and polluted environments^{15,16}. For most of the events observed at the ENA site, continuous growth of nucleation mode particles was observed despite substantial shift in wind direction, suggesting regional scale events. As these regional-scale events occur over relatively large spatial scales, they are unlikely due to local point sources. The relatively large scale of these new particle formation events is supported by the aircraft measurements (Fig. 1C and Supplementary Fig. 7C). Second, the nucleation mode particles are also unlikely due to local sources on Graciosa Island. Early studies show that new particles may form from iodine oxides in coastal regions. Such NPF coincides with low tide in the presence of solar radiation¹⁷. For the growth events observed at the ENA site, the appearance of the nucleation mode particles does not correlate with low tide, indicating a different mechanism. For example, nucleation mode particles were first observed at ~ 15:00 UTC on 16 February 2018, coinciding with the high tide on that day. In addition, the observed particle growth rates between 10 and 35 nm were mostly 1 nm h⁻¹ or lower (Supplementary Table 1). Earlier studies¹⁸⁻²¹ showed that particle growth rate decreases with decreasing particle diameter below ~ 40 nm, likely due to the stronger Kelvin effect of smaller particles. Therefore, the growth rates below 10 nm are expected to be even lower than 1 nm h⁻¹. The attribution of the nucleation mode particles to NPF over open ocean is further based on the fact that this slow growth rate is insufficient to grow particles from 1 nm to several tens of nanometers in less than 1-2 hours (i.e., the maximum transit time from the shore/tidal region to the ENA site). Third, the nucleation mode particles and particle growth are also unlikely due to emissions from surrounding Azores islands, as evidenced by the analysis of air mass backward trajectories. We calculated 3-day back-trajectories of air masses arriving at 100 m above the ground level at the ENA site, for each hour during the observed events. For over 50% of the events, no trajectories intersected with any of the Azores islands (Supplementary Fig. 10a). For only 12% of the events, more than 50% of the hourly air mass trajectories during the event passed over one or more of the Azores islands (Supplementary Fig. 10a). However, the particle growth rate exhibits no dependence on the fraction of hourly trajectories that passed over the Azores islands during the event (Supplementary Fig. 10b), indicating that observed nucleation mode particles and particle

growth are unlikely due to biogenic emissions or anthropogenic activities on surrounding Azores islands.”

Figure R1 was added as Supplementary Fig. 10 in the revised manuscript.

455 **Comment:**

• The slow growth rate from 1 - ~10 nm referred to on line 378 seem to be the crucial argument for why the growth events seen on the ground site are from particles nucleated over the open ocean, and therefore critical for claiming that these particles form CCN. In supplementary table 1 the growth rate for each observed event is given – how is that calculated? And what size range does it apply to? If the particles are only observed on the island above ~10 nm, how can we infer that these growth rates apply to particles < 10 nm? I believe there may also be a missing word in this sentence which many need to be corrected.

460 **Response:**

Here the growth rates are derived from linear fitting of the time series of the particle mode diameters ($D_{p,n}$) between 10 and 35 nm (i.e., GR₁₀₋₃₅). This is now clarified in Supplementary Table 1. Earlier studies show that particle growth rate generally decreases with decreasing particle diameter below ~ 40 nm¹⁸⁻²¹. Such decrease is partially attributed to the stronger Kelvin effect and thus higher surface equilibrium vapor pressure of smaller particles (e.g., ref.¹⁹). Therefore, the derived growth rates between 10 and 35 nm in Supplementary Table 1 should represent upper limits of particle growth rate below 10 nm, and the time required for particles to grow from ~1 nm to several tens of nanometers is even longer than that estimated based on GR₁₀₋₃₅ (i.e., > 1-2 hours).

470 We’ve modified the description accordingly to (line 493):

“In addition, the observed particle growth rates between 10 and 35 nm were mostly 1 nm h⁻¹ or lower (Supplementary Table 1). Earlier studies¹⁸⁻²¹ showed that particle growth rate decreases with decreasing particle diameter below ~ 40 nm, likely due to the stronger Kelvin effect of smaller particles. Therefore, the growth rates below 10 nm are expected to be even lower than 1 nm h⁻¹. The attribution of the nucleation mode particles to NPF over open ocean is further based on the fact that this slow growth rate is insufficient to grow particles from 1 nm to several tens of nanometers in less than 1-2 hours (i.e., the maximum transit time from the shore/tidal region to the ENA site).”

480 **Comment:**

485 • On line 381 the continuous growth of nucleation mode particles is claimed as evidence for this being a regional phenomenon. Is this because there is evidence of a supply of small particles over a long period of time? This point needs some clarification.

Response:

Please see our responses to comment #1 in this section (line 387-453 in this file).

490 **Comment:**

• The raman lidar profiles on event days are said to show “reduced aerosol extinction in the upper decoupled layer” (line 397) – supplementary table 1 shows only the data for event days. Statistics from non-event days should be provided to back up this claim. It would be helpful to also provide easy summary statistics from events days to make the comparison directly.

495 **Response:**

We thank Dr. Williamson for this suggestion. We first used the Balloon-Borne Sounding System (SONDE) data at the ENA site²² to determine the MBL height and structure (i.e., well-mixed or decoupled). The Raman lidar profiles corresponding to the time of the sounding measurements are examined. Only extinction data characterized as “aerosol” type are included in the analysis to 500 exclude the large extinction from clouds²³. To exclude the influence of extreme values, we represent the aerosol extinction for each case using the median value in the upper decoupled layer (when the boundary layer was decoupled) or in the upper 1/3 of a well-mixed MBL. Figure R2 shows the statistics of the aerosol extinction for the cases of (1) well mixed boundary layers, (2) decoupled boundary layers outside the event periods, and (3) decoupled boundary layer 505 during the events. Aerosol extinction in the decoupled upper layer is generally lower than the upper 1/3 layer of the well-mixed MBL. In addition, for decoupled MBL, the aerosol extinction in the upper decoupled layer on event days is statistically lower than that of non-event days, indicating reduced condensation and coagulation sinks.

We also note that a reduced extinction (i.e., condensation and coagulation sinks) alone is 510 necessary but not sufficient for NPF to occur. Temperature, precursor concentration, and cloud condition also likely play an important role.

515 **Figure R2. Statistics of median aerosol extinction in the top 1/3 of well mixed MBLs, upper decoupled layer outside of the growth event periods, and upper decoupled layer during the events.**

Related discussion has been added in the revised manuscript as (line 524):

520 “For all 35 growth events when Raman Lidar measurements are available and with signals above the detection limit (Supplementary Table 1), the retrieved vertical profiles show reduced aerosol extinction in the upper decoupled layer, indicating low surface area concentration that is favorable to NPF, as shown in the cases presented earlier. For decoupled MBLs, the aerosol extinction in the upper decoupled layer on event days has a median value of 0.04 km⁻¹, and is generally lower than that of non-event days (median value of 0.05 km⁻¹), indicating reduced condensation and coagulation sinks. In addition, extinction in the decoupled upper layer is generally lower than that in the upper 1/3 layer of a well-mixed MBL (median value of 0.07 km⁻¹).”

525

Comment:

530 • Supplementary figure 8b shows a growth event and is cited as evidence of newly formed particles growing to CCN sizes. However, the particles here appear not to grow larger than ~ 40 nm, which is generally considered too small to act as a CCN. Can the authors explain their reasoning in assuming these particles do grow to CCN sizes before being lost to sinks? They

claim in line 452 that anomalously strong updrafts could remove enough particles around 50 nm in diameter that they do not show up in the growth bananas. Can the authors explain why this is the case here, but other ground site observations of new particle formation and growth see particles well beyond 50 nm? Furthermore, the authors show the Hoppel minimum here to be at 80 nm (line 458), which also suggests that we should still see more 50 nm particles than are present in the size distribution. I'm struggling to follow this part of the argument and would appreciate some further explanation.

540 **Response:**

One main reason that the particles appear not to grow larger than ~40-50 nm at the ENA site is because cloud processing strongly modifies the shape of aerosol size distribution in the MBL. New particle formation and growth beyond 50 nm were indeed reported in previous studies, which are essentially all based on observations at continental sites, where the influence of cloud processing on the aerosol size spectrum is likely weaker. The Hoppel minimum of 80 nm is based on the annual average²⁴. During the post-cold-frontal periods when nucleation mode particles and particle growth were observed (Supplementary Table 1), the Hoppel minimum averaged 64 ± 7 nm. It is worth noting that the Hoppel minimum represents the “average” threshold size of CCN. There is a distribution of updraft velocity for cloud formation inside MBL. Particles with diameters smaller than the Hoppel minimum can activate and become cloud droplets in stronger-than-average updrafts. Once activated, in-cloud production of sulfate and organics increases the amount of the solute in the droplets. If these cloud droplets are not removed by wet scavenging, they can return to particles with increased diameters above the Hoppel minimum upon evaporation, and readily serve as CCN during the subsequent cloud formation. Martin et al.²⁵ estimated the peak supersaturation (i.e., supersaturation near the cloud base where droplets are activated) of mid-latitude stratocumulus clouds using droplet closure, and the results show a considerable fraction of the peak supersaturations greater than 0.4%, and the peak supersaturation can reach as high as 0.7%. For ammonium sulfate particles with a κ value of 0.61, particles as small as 50 nm can be activated into cloud droplets under a supersaturation of 0.4%. Therefore, while some of the particles grow and reach the Hoppel minimum, we do not observe the growth of the *nucleation mode size* to the Hoppel minimum because (1) a substantial fraction of the particles smaller than the Hoppel minimum are activated in stronger-than-average updrafts and subsequently “jump” to the accumulation mode as a result of cloud processing, and (2) the particle mode size is expected to be substantially below the Hoppel minimum, which represents the minimum between the nucleation/Aitken and accumulation modes.

We've included this discussion in the revised Methods as (line 466):

570 “The time of these events and the corresponding cold front passages is illustrated in
Supplementary Fig. 8A. An example of the growth events is shown in Supplementary Fig.
8B (see more discussion in “Contribution of condensational growth of newly formed
particles to MBL CCN” section).”

And in line 587 as:

575 “It is worth noting that we did not directly observe the growth of particle mode diameter to
CCN size range or the average Hoppel minimum (e.g., Supplementary Fig. 8B) because
cloud processing strongly modifies the shape of the aerosol size distribution in the MBL. At
the ENA site, the Hoppel minimum is typically around 80 nm based on the annual average²⁴.
During the post-cold-frontal periods when nucleation mode particles and particle growth
were observed (Supplementary Table 1), the Hoppel minimum averaged 64 ± 7 nm. The
Hoppel minimum represents the “average” threshold size of CCN. Updraft velocity for
580 cloud formation inside the MBL exhibits a range of values. Particles with diameters below
the Hoppel minimum can activate and become cloud droplets in stronger-than-average
updrafts. Once activated, in-cloud production of sulfate and organics increases the amount
of the solute in the droplets more efficiently than condensational growth. If these cloud
droplets are not removed by wet scavenging, they can return to particles with increased
585 diameters above the Hoppel minimum upon evaporation, and readily serve as CCN during
the subsequent cloud formation. This process leads to the typical bimodal aerosol size
distribution in the MBL²⁶. Martin et al.²⁵ estimated the peak supersaturation (i.e.,
supersaturation near cloud base where droplets are activated) of mid-latitude stratocumulus
clouds using droplet closure, and the results show a considerable fraction of the peak
590 supersaturations greater than 0.4%, and the peak supersaturation can reach as high as
0.7%. For ammonium sulfate particles with a κ value of 0.61, particles as small as 50 nm can
be activated into cloud droplets under a supersaturation of 0.4%. Therefore, while some of
the particles grow and reach the Hoppel minimum, we do not observe the growth of the
nucleation mode size to the Hoppel minimum because (1) a substantial fraction of the
595 particles smaller than the Hoppel minimum are activated in stronger-than-average updrafts
and subsequently “jump” to the accumulation mode as a result of cloud processing, and (2)
the particle mode size is expected to be substantially below the Hoppel minimum, which
represents the minimum between the nucleation/Aitken and accumulation modes.”

600 **Comment:**

- The simulated growth mentioned in line 472 – how does this compared with the measured growth? Could this be shown in a figure? I’m also unclear about why in the simulations particles

grow to 80 nm in clear air, but in the observed size distributions they do not grow this large. To me that would indicate a difference between the simulations and observations that brings into question the quote 50% and 70% of CCN (line 487).

Response:

Sorry for the confusion. Unfortunately, we don't have direct measurements of the condensing species at the ENA site. In L471-472, we meant that we derived the growth rate (GR) between ~35 nm and 80 nm based on the observed growth rate below ~35 nm, as GR is a function of particle diameter (D_p) at the same vapor pressure of nonvolatile condensing species. The growth rate, i.e. rate of diameter change of a particle due to condensation, can be calculated as²:

$$GR(D_p) = \frac{dD_p}{dt} = \frac{4D_i M_i}{RT \rho_p} \cdot \frac{f(Kn, \alpha)}{D_p} \cdot (p_i - p_{eq,i}) \quad (R4a)$$

where

$$f(Kn, \alpha) = \frac{0.75\alpha(1 + Kn)}{Kn^2 + (1 + 0.283\alpha)Kn + 0.75\alpha} \quad (R1b)$$

where D_i is the mass diffusivity of condensate i in air, M_i is its molecular weight, R is the gas constant, T is temperature in K, ρ_p is the particle density. p_i and $p_{eq,i}$ are the vapor pressures of condensate i in the bulk gas-phase and at the aerosol surface, respectively. For a nonvolatile condensate like H_2SO_4 , $p_{eq,i}$ is zero³. Kn is the Knudsen number defined as $2\lambda_{mfp}/D_p$, where λ_{mfp} is the mean free path. The term $f(Kn, \alpha)$ is the correction due to non-continuum effects (depending on Kn) and imperfect surface accommodation (depending on mass accommodation coefficient α), and is estimated by the Fuchs-Sutugin approach².

The above equation shows that at the same condensate vapor pressure, the size dependence of GR originates from the term $f(Kn, \alpha)D_p^{-1}$. When D_p is below 35 nm, this dependence is negligible and GR can be estimated from a linear fitting of the time series of $D_{p,n}$. In the size range of 35 to 80 nm, GR is derived assuming the same condensate vapor pressure, and with the size dependence taken into account using the term $f(Kn, \alpha)D_p^{-1}$.

As described above, we did not observe the growth of the nucleation mode size to the Hoppel minimum because a substantial fraction of the particles smaller than the Hoppel minimum are activated in stronger-than-average updrafts and subsequently “jump” to the accumulation mode as a result of cloud processing. In essence, a substantial fraction of the particles become CCN when reaching a smaller size than the Hoppel minimum. Therefore, the contribution by the

growth of newly formed particles to CCN may be even higher than the calculated value, which represents the number of particles that survive the coagulation and reach 80 nm.

635 **Comment:**

- Some more stats are needed on the CCN contribution from NPF e.g. the range or standard deviation to give some idea of how much this varies.

Response:

640 Following Dr. Williamson's suggestions, we've included more statistics in the main text as (line 197):

“On average, the growth of nucleation mode particles, which originate from the upper decoupled layer, contributes $143 \pm 118 \text{ cm}^{-3}$, $50\% \pm 22\%$ of the total CCN concentration. The contribution potentially reaches $70\% \pm 20\%$ under the cleanest conditions when the pre-existing CCN concentration is below 60 cm^{-3} (Fig. 4).”

645

Comment:

There are some points that need clarification regarding how widespread and important this method of CCN production is:

650 • The paper claims that because favorable conditions for NPF are widespread, and growth to CCN sizes are observed from Graciosa island, that production of CCN from this NPF is also widespread. As the paper stands, it is not clear to me that the possibility that condensable vapors associated with the island (e.g. coastal, anthropogenic, terrestrial-biogenic) contribute substantially to the growth, has been sufficiently ruled out for this claim to be made. Can the authors explain this further or supply more evidence?

655 **Response:**

See our response in last section (line 387-453 in this file).

Comment:

660 • Line 38: ref 2 is cited to say that CCN concentrations are responsible for a significant fraction of variability in MBL radiative effects, but I believe this reference actually says that CCN effects on albedo variance is small when compared with factors such as natural meteorological variability. While the authors correctly cite ref. 3, which supports their argument that CCN are

important here, they need to address the fact that there are conflicting arguments in the literature on the importance of CCN variation for albedo modification in the MBL.

665 **Response:**

Thanks for pointing this out. George and Wood²⁷ show that the variations in cloud droplet concentrations (N_d) constitute ~ 5 -10% of the day to day temporal variance in albedo over the southeast Pacific region. Although this is smaller than the contribution from the variations in cloud fraction and cloud liquid water path, which dominate, it is important to bear in mind that the day to day variability in cloud fraction and condensate is likely influenced by variations in N_d . Rosenfeld et al.²⁸ assume that the cloud cover variability is significantly affected by variability in N_d through indirect effects. Mesoscale morphology in LES simulations^{29,30} also shows large sensitivity to N_d . Because of these significant impacts of cloud microphysical properties on cloud macrophysics, N_d is likely responsible for a larger fraction of variability in MBL radiative effects than the Twomey analysis of George and Wood indicates. One can think of the two cited papers as representing the minimum vs. maximum contributions of N_d to MBL radiative effects. In either case, substantial variability in the radiative effects of marine low clouds can be attributed to the concentration of cloud condensation nuclei (CCN). We have modified the sentence to (line 38):

680 “Substantial variability in their radiative effects is attributed to the concentration of cloud condensation nuclei (CCN)^{27,28}.”

Comment:

685 • The contribution of newly formed particles to CCN is stated as 50-70% on line 189. My understanding from the text is that this is instantaneous i.e. at the end of the NPF + growth event and local. To understand the importance of MBL CCN we need to know not only the instantaneous contribution to CCN locally, but also some estimation of the contribution globally over longer timescales. It would be very helpful if the authors could tie this together with what they mention about the conditions for this phenomenon being both widespread and frequent to estimate also the average contribution annually and globally (or as a function of the total annual MBL CCN). While there will be large uncertainties in this estimate, it would give context for the importance of this new mechanism. I am also struggling to understand where the 50% and 70% quoted here are calculated. To me it is not clear from fig 4.

Response:

695 We thank Dr. Williamson for this comment. Yes, the contribution represents the contribution at the end of the NPF and growth events. The relative frequency of occurrence of cold fronts is

generally 5% to 30% over the mid-latitude oceans³¹. Within the cold air outbreak regions, the occurrence probability of broken clouds (i.e., the conditions for NPF to occur) is ~45%³².

700 Together, we estimate the NPF in the upper part of the MBL to occur ~2.5% to ~14% of the time over mid-latitude ocean, consistent with the frequency of 12% observed at the ENA site. The frequency of 12% is estimated using the ratio of the total duration of the observed events to the total sampling time (Supplementary Table 1). This frequency has a similar magnitude as the frequency of continental NPF events of 10% ~ 30%³³. In addition, the oceans cover a larger area than the lands in the mid-latitudes. It is important to note that CCN generated from the growth of
705 newly formed particles continue to contribute to the CCN population and the formation of marine low clouds after the NPF and growth events end. Before the onset of the next NPF and growth event, the contribution may gradually decrease due to scavenging and the production of MBL CCN through other processes. To understand the annual and global average contribution requires systematic simulations using a global model that represents other sources of CCN
710 besides the MBL NPF as well as the sinks of CCN, which are beyond the scope of this manuscript and will be the focus of a future study.

We've added this information into the manuscript as (line 197):

“On average, the growth of nucleation mode particles, which originate from the upper decoupled layer, contributes $143 \pm 118 \text{ cm}^{-3}$, $50\% \pm 22\%$ of the total CCN concentration.
715 The contribution potentially reaches $70\% \pm 20\%$ under the cleanest conditions when the pre-existing CCN concentration is below 60 cm^{-3} (Fig. 4), suggesting NPF in the MBL is likely an important source that helps replenish aerosol and CCN populations in the pristine marine environment. Nucleation mode particles were also frequently observed in the remote MBL following the passage of cold fronts during previous research cruises in the mid-latitude
720 Pacific Ocean, and in the southern Pacific and Indian Oceans^{34,35}. These nucleation mode particles are conventionally attributed to the entrainment of new particles formed in the FT³⁴⁻³⁶. The similar observations of nucleation mode particles in the MBL following the passage of cold fronts in different regions also indicate that the findings presented herein may be applied more generally to mid-latitude oceans. The relative frequency of occurrence
725 of cold fronts is generally 5% to 30% over the mid-latitude oceans³¹. Within the cold air outbreak regions, the occurrence probability of broken clouds (i.e., the conditions for NPF to occur) is ~45%³². Together, we estimate the NPF in the upper part of the MBL occurs ~2.5% to ~14% of the time over mid-latitude oceans, consistent with a frequency of 12% based on the observations at the ENA site (Supplementary Table 1). This frequency has a
730 similar magnitude as the frequency of continental NPF events of 10% ~ 30%³³, suggesting that new particle formation in the upper MBL likely plays an important role in the budget of MBL CCN.”

Comment:

735 • Line 116 states that upper MBL NPF occurs over a wide area. This could be better supported if a map of all profiles measured where NPF was and wasn't observed along with cloud height, Stot etc.

Response:

740 We meant that the NPF occurs in different geographic areas of mid-latitude ocean (i.e., not specific to the Azores area) as the NAAMES flight was carried out in the North Atlantic. We have rephrased the sentence to (line 121):

“suggesting that NPF occurs in the upper MBL over different geographic areas of mid-latitude oceans (i.e., is not specific to the Azores).”

745 **Comment:**

In the main data, the total surface area is used as a proxy for the coagulation sinks. I have a few questions about this:

750 • Why did the authors choose to use Stot instead of calculating the coagulation sink for e.g. 3 nm particles, from the size distribution? If they wish to use this proxy instead of a calculated sink, a supplementary figure showing its close correlation to the coagulation sink for relevant sizes would make this choice more justified.

Response:

755 We use the S_{tot} to represent both the condensation sink of nucleation precursors and the coagulation sink of newly formed particles. We use the surface area as a surrogate for the condensation and coagulation sinks as it is easier to compare with the results from earlier studies of new particle formation in marine environments^{34,35}. To more accurately quantify the coagulation and condensation sinks caused by the pre-existing aerosols, we examined the correlations among coagulation sink of 3 nm particles, condensation sink of H₂SO₄, and the aerosol surface area during the entire ACE-ENA campaign. The coagulation sink is derived using the size distributions measured by the FIMS, PCASP, and FCDP onboard the G-1 aircraft, covering the size range from 10 nm to 50 μm. The coagulation coefficient is calculated using the equation covering the free-molecular, transition, and continuum regimes. As shown in the figures below, in general, the coagulation sink, condensation sink, and aerosol surface area show strong linear relationships, in agreement with the result from Williamson *et al.*⁸ Therefore, it is reasonable to use S_{tot} as a proxy of the condensation and coagulation sinks in this study.

760

765

Figure R3. Correlations among coagulation rate, condensation rate, and aerosol surface area during the ACE-ENA campaign. Relationships among the condensation rate of H₂SO₄ molecules onto existing particles, the coagulation rate between 3-nm particles and all other particles, and the total aerosol surface area from 10 nm to 50 μm.

770

The discussion above has been included in the revised manuscript (line 357) as:

“In this study, the total surface area concentration (S_{tot}), including that of particles and possible hydrometeors, generally shows strong linear relationships with the condensational rate of H₂SO₄ molecules and the coagulation rate of 3 nm particles, in agreement with the results from Williamson et al¹⁹. Therefore, S_{tot} is used here as a surrogate for both the condensational sink of nucleation precursors and coagulation sink of newly formed particles.”

775

780 **Comment:**

• In the methods section (Line 284) the authors mention that S_{tot} is calculated for particles larger than 10, sometimes above 100 nm, and sometimes using size distributions from the LAS instrument instead of the FIMS. While larger particles generally dominate the total coagulation sink, during nucleation events in clean conditions, the sink to smaller particles can become important. Some characterization of how much this could influence the results (e.g. using simulated nucleation events under conditions similar to the observations) would remove questions about biases in the data due to these unavoidable differences in instrumentation. A 3 – 10 nm size bin, using the CPC data could even be included in the calculated S_{tot} or coagulation sink this analysis shows the contribution to the sink from this size range to be important.

785

790 **Response:**

The FIMS was operated downstream of the thermal denuder during short periods during the ACE-ENA flights (typically ~ 20 min, less than 10% of the flight time). Figure 1 and relevant supplementary figures show vertical profiles of S_{tot} based on measurement periods when FIMS was not sampling downstream of the thermal denuder. Therefore, S_{tot} during ACE-ENA flights is derived from the size distributions ranging from 10 nm to 50 μm . Following the suggestion of Dr. Williamson, we estimated the upper limit of the contribution from particles below 10 nm to the coagulation sink by treating all the particles between 3 and 10 nm with a uniform size of 10 nm. During the entire ACE-ENA campaign, the upper limit of this contribution is $0.2 \pm 0.8\%$. During the NPF cases shown in this manuscript, the contribution is $0.8 \pm 1.5\%$, with a maximum value of 6.25%, which occurred inside the NPF region. For the NAAMES campaign, due to the low time resolution of the SMPS measurements, the vertical profile of S_{tot} is derived using the size distribution from 100 nm to 5 μm measured by the Laser Aerosol Spectrometer and Aerodynamic Particle Sizer. Using the average size distribution measured by the SMPS, we estimated that the contribution of particles below 100 nm to the total coagulation sink is $14.9 \pm 3.3\%$ in the NPF region. For this study, the coagulation sink is largely dominated by aerosols above 100 nm due to the relative abundance of accumulation-mode and sea-spray aerosols in the relevant altitude range (i.e., below ~ 3000 m). We have included the discussion in the manuscript (line 349) as:

“For measurements onboard G-1 during ACE-ENA, S_{tot} was calculated by integrating the combined surface area-size distributions measured by the FIMS, PCASP, and FCDP from 10 nm to 50 μm (Supplementary Fig. 3) assuming spherical particles. As we do not have aerosol size distribution below 10 nm, the upper limit of the contribution from particles between 3 and 10 nm to the coagulation sink is calculated by treating all particles between 3 and 10 nm with a uniform diameter of 10 nm. During the entire ACE-ENA campaign, the upper limit of this contribution is $0.2 \pm 0.8\%$. During the NPF cases shown in this manuscript, the contribution is $0.8 \pm 1.5\%$, with a maximum value of 6.25%, which occurred inside the NPF region. For the NAAMES campaign, due to the low time resolution of the SMPS measurements, the vertical profile of S_{tot} is derived using the size distribution from 100 nm to 5 μm measured by the Laser Aerosol Spectrometer and Aerodynamic Particle Sizer. Using the average size distribution measured by the SMPS, we estimated that the contribution of particles below 100 nm to the total coagulation sink is $14.9 \pm 3.3\%$ in the NPF region.”

825 *Minor comment:*

Finally, a few smaller unrelated questions:

Comment:

830 • The authors mention that only clear-sky data are used to avoid inlet artifacts. How are in-cloud and out-of-cloud time periods identified. A strong explanation this classification seems essential since we are seeing large numbers of small particles in the vicinity of clouds and the community will be keen to rule out any possibility of these being artifacts from measuring inside clouds.

Response:

835 The in-cloud time periods are identified based on the liquid water content derived from droplet spectra measured by the fast cloud droplet probe (FCDP) deployed onboard G-1 during ACE-ENA or the cloud droplet probe (CDP) deployed onboard C-130 during NAAMES. A threshold of 0.001 g m^{-3} was used to identify the period of sampling inside clouds. In addition, measurements collected during, two seconds before, and two seconds after the in-cloud sample periods were excluded to eliminate the possible impact of cloud droplet shattering on the particle measurements. The description has been added to the methods section (line 380):

840 “The in-cloud time periods are identified based on the liquid water content derived from droplet spectra measured by the FCDP deployed onboard G-1 during ACE-ENA or the cloud droplet probe deployed on C-130 during NAAMES. A threshold of 0.001 g m^{-3} was used to identify the periods of sampling inside clouds. In addition, measurements collected during, two seconds before, and two seconds after the in-cloud sample periods were
845 excluded to eliminate the possible impact of cloud droplet shattering on the particle measurements.”

Comment:

850 • I believe a small number technical correction is needed in line 126 - unit should be μm^2 instead of μm^{-2}

Response:

The unit has been corrected and we have checked all units throughout the manuscript.

Comment:

855 I believe this paper, once the issues mentioned above are addressed, will influence thinking in modeling marine clouds, understanding aerosol sources and the importance of new particle

formation. It challenges the long-held view that sinks are too high in the MBL for NPF to be an important contributor to CCN here.

References

- 860 Williamson, C.J., Kupc, A., Axisa, D. et al. A large source of cloud condensation nuclei from
new particle formation in the tropics. *Nature* 574, 399–403 (2019).
<https://doi.org/10.1038/s41586-019-1638-9>
Signed: Christina J Williamson

Response:

- 865 Again, we thank Dr. Williamson for her thoughtful and constructive comments. We also thank
her for recognizing the importance of this work.

870

Reviewer #3 (Remarks to the Author):

Comment:

875 This MS reports on observations of new particle formation in North Atlantic marine boundary layer (MBL), more specifically in the upper decoupled layer, preceded by the passage of a cold front. Previously it has been thought that secondary new particles observed in MBL are formed in free troposphere (above upper MBL layer) and mixed with MBL. Results that NPF occur in MBL also and not only in FT is important for proper description of NPF in models.

880 Explanation of observed NPF which is not caught by the models is much lower aerosol surface area in the upper MBL than assumed in the models. This work certainly should help improving models and in that sense deserves to be published. However, I do not know, if the result that NPF occurs in the atmospheric layer (upper decoupled MBL) where it was not believed to be frequent rather than (just) in FT above the upper decoupled MBL is remarkable in the eyes of wider audience or would these results suit better for a more specific journal.

885 While the key results are based on observations made during few tens of Gulfstream flights, scaling to wider geographical areas seem justified in the light of year-long measurements in southern Atlantic previous research cruises in the Pacific and Indian Oceans where new particles were frequently observed in similar meteorological situations (passage of cold front). Thus the conclusions are scientifically sound.

890 The mechanism of NPF is speculative. Concurrent vapour measurements would have added much to the study. Authors suggest DMS oxidation products (sulphuric and methane sulphonic acid) to be responsible on particle formation which is likely a correct assumption. However, without trace gas / vapour data, the mechanism, including the role of ammonia etc. bases and air ionisation remain speculative.

895 **Response:**

We thank the reviewer for the thoughtful and constructive comments. As the reviewer pointed out, the main result of this study is that new particle formation occurs in the upper part of the MBL instead of the FT as previously thought. We think this is an important result as particles formed in the MBL can readily grow to CCN size due to relatively high precursor concentrations
900 (e.g., DMS) and influence marine low clouds. In contrast, particles formed in the FT need to be vertically transported into the MBL through entrainment (a relatively slower process) before they can grow and/or influence the properties of marine low clouds.

Following the suggestion of the reviewer, we have added the vertical profiles of DMS mixing ratio during the new particle formation events in Fig. 1 and relevant supplementary figures. In
905 general, the vertical profiles of DMS resemble those of water vapor, showing elevated mixing

ratios in both surface mixed layer and upper decoupled layer compared to that in the free troposphere. This is expected as DMS originates from the ocean surface. The elevated DMS mixing ratio in the upper decoupled layer of the MBL suggests that oxidation products of DMS including H₂SO₄ likely contribute to the new particle formation observed.

910 The gas-phase DMS mixing ratio was measured by a PTR-ToF-MS during the NAAMES flights. During ACE-ENA, the mixing ratio of DMS was measured by an Ionicon quadrupole PTR-MS onboard the G-1. Because the quadrupole PTR-MS cycled through a number of pre-selected m/z values, the duty cycle of the PTR-MS for individual species is lower than that in the PTR-ToF-MS, leading to reduced signal to noise ratio. In addition, given the unit mass resolution of the
915 quadrupole PTR-MS, isobaric interference may also lead to a positive bias in the measured DMS mixing ratio. Here we only use the vertical profile of DMS qualitatively to show the relative abundance of DMS in the upper part of MBL. The conclusion of this study is not based on the absolute values of the DMS mixing ratio, and therefore is not affected. The above description and discussion have been included in the Results and discussion (line 80):

920 “The ion signal at m/z 63 measured by a Proton-Transfer-Reaction Mass Spectrometer (PTR-MS) suggests that vertical profile of dimethyl sulfide (DMS) is similar to that of water vapor, showing elevated mixing ratio in both surface mixed layer and upper decoupled layer compared to that in the free troposphere (Fig. 1B). This is expected as DMS originates from the ocean surface.”,

925 and Methods section as (line 267):

“An Ionicon quadrupole high-sensitivity Proton-Transfer-Reaction Mass Spectrometer (abbreviated as PTR-MS thereafter) was used to measure the mixing ratios of selected gas-phase VOC including DMS. Because the quadrupole PTR-MS cycled through a number of pre-selected m/z values, the duty cycle for each m/z is limited, leading to a reduced signal to
930 noise ratio. In addition, given the unit mass resolution of the quadrupole PTR-MS, isobaric interference may also lead to a positive bias in the measured DMS mixing ratio. Here we only use the vertical profile of the ion signal at m/z 63 to qualitatively infer the relative abundance of DMS in the upper part of MBL. The conclusion of this study does not depend on the absolute values of the DMS mixing ratio.”

935 We agree with the reviewer that DMS oxidation products (sulfuric and methane sulfonic acid) are likely responsible for the new particle formation observed. We also agree that without other trace gas data, we will not be able to pinpoint the mechanism and the roles of bases (e.g., ammonia) and air ionization, which will be a focus of a future field campaign/study.

References

- 940 1 Veres, P. R. *et al.* Global airborne sampling reveals a previously unobserved dimethyl sulfide oxidation mechanism in the marine atmosphere. *Proceedings of the National Academy of Sciences* **117**, 4505-4510 (2020).
- 2 Seinfeld, J. H. & Pandis, S. N. *Atmospheric chemistry and physics: from air pollution to climate change*. (John Wiley & Sons, 2016).
- 945 3 Pandis, S. N., Russell, L. M. & Seinfeld, J. H. The relationship between DMS flux and CCN concentration in remote marine regions. *Journal of Geophysical Research: Atmospheres* **99**, 16945-16957, doi:10.1029/94JD01119 (1994).
- 4 Clarke, A. D. Atmospheric nuclei in the Pacific midtroposphere: Their nature, concentration, and evolution. *Journal of Geophysical Research: Atmospheres* **98**, 20633-20647, doi:10.1029/93jd00797 (1993).
- 950 5 Clarke, A. *et al.* Particle nucleation in the tropical boundary layer and its coupling to marine sulfur sources. *Science* **282**, 89-92 (1998).
- 6 Peter, J. R. *et al.* Airborne observations of the effect of a cold front on the aerosol particle size distribution and new particle formation. *Quarterly Journal of the Royal Meteorological Society* **136**, 944-961 (2010).
- 955 7 Quan, J. *et al.* Anthropogenic pollution elevates the peak height of new particle formation from planetary boundary layer to lower free troposphere. *Geophysical Research Letters* **44**, 7537-7543 (2017).
- 8 Williamson, C. J. *et al.* A large source of cloud condensation nuclei from new particle formation in the tropics. *Nature* **574**, 399-403 (2019).
- 960 9 Takegawa, N. & Sakurai, H. Laboratory evaluation of a TSI condensation particle counter (model 3771) under airborne measurement conditions. *Aerosol Science and Technology* **45**, 272-283 (2011).
- 10 Kulmala, M. *et al.* Formation and growth rates of ultrafine atmospheric particles: a review of observations. *J Aerosol Sci* **35**, 143-176 (2004).
- 965 11 Dal Maso, M. *et al.* Formation and growth of fresh atmospheric aerosols: eight years of aerosol size distribution data from SMEAR II, Hyytiala, Finland. *Boreal environment research* **10**, 323 (2005).
- 12 Kristensson, A. *et al.* Characterization of new particle formation events at a background site in Southern Sweden: relation to air mass history. *Tellus B: Chemical and Physical Meteorology* **60**, 330-344, doi:10.1111/j.1600-0889.2008.00345.x (2008).
- 970 13 Hussein, T. *et al.* Time span and spatial scale of regional new particle formation events over Finland and Southern Sweden. *Atmos. Chem. Phys.* **9**, 4699-4716, doi:10.5194/acp-9-4699-2009 (2009).

- 975 14 Pierce, J. R. *et al.* Nucleation and condensational growth to CCN sizes during a sustained
pristine biogenic SOA event in a forested mountain valley. *Atmos. Chem. Phys.* **12**, 3147-
3163, doi:10.5194/acp-12-3147-2012 (2012).
- 15 Dunn, M. J. *et al.* Measurements of Mexico City nanoparticle size distributions:
Observations of new particle formation and growth. *Geophysical Research Letters* **31**,
980 doi:10.1029/2004gl019483 (2004).
- 16 Wiedensohler, A. *et al.* Rapid aerosol particle growth and increase of cloud condensation
nucleus activity by secondary aerosol formation and condensation: A case study for
regional air pollution in northeastern China. *Journal of Geophysical Research:*
Atmospheres **114**, doi:10.1029/2008jd010884 (2009).
- 985 17 O'Dowd, C. D. *et al.* Coastal new particle formation: Environmental conditions and
aerosol physicochemical characteristics during nucleation bursts. *J Geophys Res-Atmos*
107, doi:10.1029/2000jd000206 (2002).
- 18 Kuang, C. *et al.* Size and time-resolved growth rate measurements of 1 to 5 nm freshly
formed atmospheric nuclei. *Atmospheric Chemistry & Physics* **12**, 3573-3589 (2012).
990 <<https://www.atmos-chem-phys.net/12/3573/2012/acp-12-3573-2012.pdf>>.
- 19 Kulmala, M. *et al.* Initial steps of aerosol growth. *Atmos. Chem. Phys.* **4**, 2553-2560
(2004).
- 20 Manninen, H. E. *et al.* Charged and total particle formation and growth rates during
EUCAARI 2007 campaign in Hyytiälä. *Atmos. Chem. Phys.* **9**, 4077-4089 (2009).
- 995 21 Winkler, P. M. *et al.* Identification of the biogenic compounds responsible for size-
dependent nanoparticle growth. *Geophysical Research Letters* **39**,
doi:10.1029/2012gl053253 (2012).
- 22 Holdridge, D., Ritsche, M., Prell, J. & Coulter, R. Balloon-borne sounding system
(SONDE) handbook. (2011).
- 1000 23 Chand, D. *et al.* Aerosol and Cloud Optical Properties from the ARM Raman Lidars: The
Feature Detection and Extinction (FEX) Value-Added Product. (DOE Office of Science
Atmospheric Radiation Measurement (ARM) Program ..., 2019).
- 24 Zheng, G. *et al.* Marine boundary layer aerosol in the eastern North Atlantic: seasonal
variations and key controlling processes. *Atmos. Chem. Phys.* **18**, 17615-17635,
1005 doi:10.5194/acp-18-17615-2018 (2018).
- 25 Martin, G. M., Johnson, D. W. & Spice, A. The Measurement and Parameterization of
Effective Radius of Droplets in Warm Stratocumulus Clouds. *Journal of the Atmospheric
Sciences* **51**, 1823-1842, doi:10.1175/1520-0469(1994)051<1823:tmapoe>2.0.co;2
(1994).

- 1010 26 Hoppel, W. A., Fitzgerald, J. W., Frick, G. M., Larson, R. E. & Mack, E. J. Aerosol size distributions and optical properties found in the marine boundary layer over the Atlantic Ocean. *Journal of Geophysical Research: Atmospheres* **95**, 3659-3686, doi:10.1029/JD095iD04p03659 (1990).
- 27 George, R. C. & Wood, R. Subseasonal variability of low cloud radiative properties over
1015 the southeast Pacific Ocean. *Atmos. Chem. Phys.* **10**, 4047-4063, doi:10.5194/acp-10-4047-2010 (2010).
- 28 Rosenfeld, D. *et al.* Aerosol-driven droplet concentrations dominate coverage and water of oceanic low-level clouds. *Science* **363**, eaav0566, doi:10.1126/science.aav0566 (2019).
- 29 Wang, H. & Feingold, G. Modeling Mesoscale Cellular Structures and Drizzle in Marine
1020 Stratocumulus. Part I: Impact of Drizzle on the Formation and Evolution of Open Cells. *Journal of the Atmospheric Sciences* **66**, 3237-3256, doi:10.1175/2009jas3022.1 (2009).
- 30 Berner, A. H., Bretherton, C. S. & Wood, R. Large-eddy simulation of mesoscale dynamics and entrainment around a pocket of open cells observed in VOCALS-REx RF06. *Atmos. Chem. Phys.* **11**, 10525-10540, doi:10.5194/acp-11-10525-2011 (2011).
- 1025 31 Fletcher, J., Mason, S. & Jakob, C. The Climatology, Meteorology, and Boundary Layer Structure of Marine Cold Air Outbreaks in Both Hemispheres*. *Journal of Climate* **29**, 1999-2014, doi:10.1175/jcli-d-15-0268.1 (2016).
- 32 Fletcher, J. K., Mason, S. & Jakob, C. A Climatology of Clouds in Marine Cold Air Outbreaks in Both Hemispheres. *Journal of Climate* **29**, 6677-6692, doi:10.1175/jcli-d-15-0783.1 (2016).
- 1030 33 Nieminen, T. *et al.* Global analysis of continental boundary layer new particle formation based on long-term measurements. *Atmos. Chem. Phys.* **18**, 14737-14756, doi:10.5194/acp-18-14737-2018 (2018).
- 34 Bates, T. S. *et al.* Processes controlling the distribution of aerosol particles in the lower
1035 marine boundary layer during the First Aerosol Characterization Experiment (ACE 1). *Journal of Geophysical Research: Atmospheres* **103**, 16369-16383 (1998).
- 35 Covert, D., Kapustin, V., Bates, T. & Quinn, P. Physical properties of marine boundary layer aerosol particles of the mid - Pacific in relation to sources and meteorological transport. *Journal of Geophysical Research: Atmospheres* **101**, 6919-6930 (1996).
- 1040 36 Pirjola, L., O'Dowd, C. D., Brooks, I. M. & Kulmala, M. Can new particle formation occur in the clean marine boundary layer? *Journal of Geophysical Research: Atmospheres* **105**, 26531-26546, doi:doi:10.1029/2000JD900310 (2000).

REVIEWERS' COMMENTS

Reviewer #1 (Remarks to the Author):

I appreciate the authors taking time to revise the manuscript and clarify my questions. I'm satisfied with the revisions and support publishing this paper.

Reviewer #2 (Remarks to the Author):

The authors have thoroughly addressed the questions and concerns I raised in my first review of the manuscript. The result is more robust and clearer than the first version. The arguments for new particle formation occurring in the marine boundary layer and contributing to CCN are convincing. The proposed mechanism is well justified. This result is of importance to the field and relevant for publication in Nature Communications. (Please see my comments in my initial review for a full explanation of the importance of this work and why it is suitable for publication in Nature Communications).

I have one remaining concern on the issue of CPC flow stability. The authors justify that the flows are stable up to 2 km altitude, but results are shown up to 4 km with statistically significant NPF between 3-4 km as strong as the NPF at the top of the MBL. This would fit with the fact that NPF has frequently been observed in the free troposphere, as the authors note in the introduction. But, given that the CPC flows may not be as stable at these altitudes, do they have confidence in their data here? It might be necessary to state that there is less confidence in the data above 2 km because of potential flow instabilities, and to reflect this somehow in the error bars. It would be helpful to point out that this result, even if less robust than the lower altitude data, does fit with previous observations.

A final minor comment. The new figure R2, the caption explains the plot as "statistics of median aerosol extinction". It is necessary to clarify what the boxes and whiskers in this plot represent e.g. interquartile and inter-decile range?

Signed: Christina J Williamson

Reviewer #3 (Remarks to the Author):

MS has improved, adding DMS measurements strengthen the message. Results are unique and important for scientific community and the to some extent reduce the speculative nature of NPF mechanism. Open questions about mechanism of NPF and the role of different vapours, which still remain, on the other hand, open new research opportunities. The main message of the MS is not anyway in detailed NPF mechanism, but the fact that NPF occurs in upper decoupled MBL where it wasn't believed to occur. I thus recommend the MS for publication.

Reviewer #1 (Remarks to the Author):

I appreciate the authors taking time to revise the manuscript and clarify my questions. I'm satisfied with the revisions and support publishing this paper.

Response:

5 We thank the reviewer for the positive comments.

Reviewer #2 (Remarks to the Author):

10 The authors have thoroughly addressed the questions and concerns I raised in my first review of the manuscript. The result is more robust and clearer than the first version. The arguments for new particle formation occurring in the marine boundary layer and contributing to CCN are convincing. The proposed mechanism is well justified. This result is of importance to the field and relevant for publication in Nature Communications. (Please see my comments in my initial review for a full explanation of the importance of this work and why it is suitable for publication in Nature Communications).

15 I have one remaining concern on the issue of CPC flow stability. The authors justify that the flows are stable up to 2 km altitude, but results are shown up to 4 km with statistically significant NPF between 3-4 km as strong as the NPF at the top of the MBL. This would fit with the fact that NPF has frequently been observed in the free troposphere, as the authors note in the introduction. But, given that the CPC flows may not be as stable at these altitudes, do they have
20 confidence in their data here? It might be necessary to state that there is less confidence in the data above 2 km because of potential flow instabilities, and to reflect this somehow in the error bars. It would be helpful to point out that this result, even if less robust than the lower altitude data, does fit with previous observations.

Response:

25 We thank Dr. Williamson for the comment on the CPC flow stability at high altitudes. In our previous response, we stressed that the flow rates were stable below the altitude of 2 km because this study focuses on the new particle formation in the marine boundary layer, which typically has a height of below ~ 2 km. The CPC flow rates were stable up to 4 km (i.e., the entire altitude range sampled during the ACE-ENA campaign), based on the same analyses described
30 in our previous response. The analyses are also discussed below.

The flow rate of CPC 3772 is controlled by a critical orifice. During the ACE-ENA flights, both the pressures downstream and upstream of the critical orifice were recorded. The pressure

ratio across the orifice (i.e., downstream pressure over the upstream pressure) indicates that the CPC flow remains critical (i.e., choked) up to 4 km above the ocean. Therefore, we are confident that the flow rate of the CPC 3772 was stable at high altitudes as well.

The sample flow rate of the CPC 3025A was not actively controlled. However, during the periods when newly formed particles were absent (based on the shape of aerosol size distribution), the number concentration ratio of the CPC 3025A to the CPC 3772 (i.e., $N_{>3\text{nm}}/N_{>10\text{nm}}$) does not change appreciably with sampling altitude up to 4 km, indicating the flow-rate stability in the CPC 3025A over the entire sampling altitude range.

A final minor comment. The new figure R2, the caption explains the plot as “statistics of median aerosol extinction”. It is necessary to clarify what the boxes and whiskers in this plot represent e.g. interquartile and inter-decile range?

Response:

The box-whisker plot is drawn for 10-, 25-, 50-, 75-, and 90-percentiles. The red circles represent the mean values.

Reviewer #3 (Remarks to the Author):

MS has improved, adding DMS measurements strengthen the message. Results are unique and important for scientific community and the to some extent reduce the speculative nature of NPF mechanism. Open questions about mechanism of NPF and the role of different vapours, which still remain, on the other hand, open new research opportunities. The main message of the MS is not anyway in detailed NPF mechanism, but the fact that NPF occurs in upper decoupled MBL where it wasn't believed to occur. I thus recommend the MS for publication.

Response:

We thank the reviewer for the positive comments.

60